# The *Drosophila* F-box protein Fbxl7 binds to the protocadherin Fat and regulates Dachs localization and Hippo signaling

**Justin A Bosch[1], Taryn M Sumabat[1], Yassi Hafezi[1], Brett J Pellock[2], Kevin D Gandhi[1], Iswar K Hariharan[1]\***

[1]Department of Molecular and Cell Biology, University of California, Berkeley, Berkeley, United States; [2]Department of Biology, Providence College, Providence, United States

**Abstract** The *Drosophila* protocadherin Fat (Ft) regulates growth, planar cell polarity (PCP) and proximodistal patterning. A key downstream component of Ft signaling is the atypical myosin Dachs (D). Multiple regions of the intracellular domain of Ft have been implicated in regulating growth and PCP but how Ft regulates D is not known. Mutations in *Fbxl7*, which encodes an F-box protein, result in tissue overgrowth and abnormalities in proximodistal patterning that phenocopy deleting a specific portion of the intracellular domain (ICD) of Ft that regulates both growth and PCP. Fbxl7 binds to this same portion of the Ft ICD, co-localizes with Ft to the proximal edge of cells and regulates the levels and asymmetry of D at the apical membrane. Fbxl7 can also regulate the trafficking of proteins between the apical membrane and intracellular vesicles. Thus Fbxl7 functions in a subset of pathways downstream of Ft and links Ft to D localization.

## Introduction

An important goal for developmental biologists is to understand how organs achieve a predictable size and shape at the end of their development. The Hippo signaling pathway has emerged as a key regulator of organ size (reviewed by *Pan, 2010*; *Halder and Johnson, 2011*; *Tapon and Harvey, 2012*). While most components of this pathway were originally discovered using genetic screens in *Drosophila*, mammalian orthologs of those genes perform similar functions. Additionally, mutations in several components of the pathway have been described in human cancers. An exciting aspect of the Hippo pathway is that its growth-regulating activity can be modulated by cell-surface proteins that are capable of binding to ligands expressed on adjacent cells. Such interactions may be especially important for achieving precise control of growth at a local level that is necessary for generating the detailed features of an organ.

Of the proteins that regulate the Hippo pathway, much research has focused on the protocadherin Fat (Ft). In addition to regulating growth, Ft also regulates planar cell polarity (PCP), oriented cell division and proximodistal patterning of appendages (reviewed in *Thomas and Strutt, 2012*; *Sharma and McNeill, 2013*) and its regulated activity therefore impacts the size and shape of organs. The Ft protein localizes to the cell membrane just apical to the adherens junctions (*Ma et al., 2003*). It has a large extracellular domain composed of 34 cadherin domains as well as 4 EGF-like domains and 2 laminin G domains (*Mahoney et al., 1991*) that binds to another large cadherin, Dachsous (Ds) (*Clark et al., 1995*), on adjacent cells (*Matakatsu and Blair, 2004*). Ft–Ds interactions are modulated by the kinase Four-Jointed (Fj), which resides in the Golgi and phosphorylates the extracellular domains of both Ft and Ds (*Ishikawa et al., 2008*; *Brittle et al., 2010*; *Simon et al., 2010*). Both Ds and Fj are expressed in gradients in *Drosophila* imaginal discs where they function in patterning the disc along a major axis (e.g., equatorial to polar or proximodistal) (*Yang et al., 2002*; *Ma et al., 2003*).

**\*For correspondence:** ikh@berkeley.edu

**Competing interests:** The authors declare that no competing interests exist.

**Reviewing editor**: Helen McNeill, The Samuel Lunenfeld Research Institute, Canada

**eLife digest** Multi-cellular organisms are made up of cells that are organized into tissues and organs that reach a predictable size and shape at the end of their development. To do this, cells must be able to sense their position and orientation within the body and know when to stop growing.

Epithelial cells—which make up the outer surface of an animal's body and line the cavities of its internal organs—connect to each other to form flat sheets. These sheets of cells contain structures that are oriented along the plane of the sheet. However, how this so-called 'planar cell polarity' coordinates with cell growth in order to build complex tissues and organs remains to be discovered.

A protein called Fat is a major player in both planar cell polarity and the Hippo signaling pathway, which controls cell growth. As such, the Fat protein appears to be crucial for controlling the size and shape of organs. Mutations in the Fat protein cause massive tissue overgrowth, prevent planar cell polarity being established correctly, and stop the legs and wings of fruit flies developing normally.

The Fat protein also plays a role in distributing another protein called Dachs—which is also part of the Hippo signaling pathway. In epithelial cells of the developing wing, Dachs is mostly located on the side of the cell that is closest to the tip of the developing wing (the so-called 'distal surface'). How Fat and Dachs work together is not understood, but it is known that they do not bind to each other directly.

Now, Bosch et al. show that in the fruit fly *Drosophila*, the Fat protein binds to another protein called Fbxl7. Flies that cannot produce working Fbxl7 have defects in some aspects of planar cell polarity and a modest increase in tissue growth. Fbxl7 seems to account for part, but not all, of the ability of Fat to restrict tissue growth. Furthermore, a lack of the Fbxl7 protein results in a spreading of Dachs protein across the apical surface—which faces out of the epithelial sheet—of epithelial cells. On the other hand, if Fbxl7 is over-expressed, Dachs is driven to the interior of each cell. Hence, a normal level of Fbxl7 protein restricts the Dachs protein to the correct parts of the cell surface.

Together, the findings of Bosch et al. show that the Fbxl7 protein is a key link between the Fat and Dachs proteins. These results also provide an understanding of how growth and planar cell polarity—two processes that are essential for normal development of all multi-cellular organisms—are coordinated.

While cadherins are known to have important functions in cell–cell adhesion, a key aspect of Ft function is its role as a signaling molecule (*Matakatsu and Blair, 2006*). Ft regulates the Hippo pathway in two ways. First, Ft influences the protein levels of Warts (Wts), a kinase that regulates the activity and subcellular location of the pro-growth transcriptional co-activator Yorkie (Yki) (*Cho et al., 2006*; *Rauskolb et al., 2011*). Additionally, mutations in *ft* disrupt the localization of Expanded (Ex), a FERM-domain protein that functions upstream of Hippo (Hpo) (*Bennett and Harvey, 2006*; *Silva et al., 2006*; *Willecke et al., 2006*), though other studies suggest Ft and Ex act in parallel (*Feng and Irvine, 2007*).

A key downstream target of Ft is the atypical myosin Dachs (D). The strong overgrowth elicited by *ft* mutations can be completely suppressed by loss of D function (*Cho et al., 2006*). Additionally, PCP defects in *ft* mutants are partially rescued by loss of D (*Mao et al., 2006*). D localizes to the apical membrane where, in cells of the wing disc, it localizes preferentially to the distal edge of the cell (*Mao et al., 2006*; *Mao et al., 2011*; *Ambegaonkar et al., 2012*; *Bosveld et al., 2012*; *Brittle et al., 2012*). In *ft* mutants, increased levels of D are observed apically and D is redistributed around the entire perimeter of the cell (*Mao et al., 2006*; *Brittle et al., 2012*). However, the overall levels of D protein are not obviously changed (*Mao et al., 2006*). It has been proposed that Ft restricts growth by negatively regulating the levels of D at the apical membrane and that it regulates the D-dependent PCP functions by maintaining D asymmetry (*Rogulja et al., 2008*).

An important gap in our current understanding of Ft function is how Ft regulates the levels and localization of D at the apical membrane. Ft does not bind to D itself, indicating that there must be one or more proteins that bind to Ft and mediate its regulation of D localization at the membrane. In an attempt to identify signaling pathways downstream of Ft, several recent studies have made

systematic deletions in the intracellular domain (ICD) of Ft (*Matakatsu and Blair, 2012*; *Bossuyt et al., 2013*; *Pan et al., 2013*; *Zhao et al., 2013*). These deletion studies implicate multiple non-overlapping regions in the ICD that differentially affect growth, PCP and organ shape, suggesting that Ft signals via multiple effector pathways. Additionally, several proteins have been shown to bind to the Ft ICD including the transcriptional repressor Atrophin/Grunge which regulates PCP (*Fanto et al., 2003*), the novel protein Lowfat that regulates Ft protein levels (*Mao et al., 2009*), and the casein kinase I protein Discs overgrown (Dco) that phosphorylates the Ft ICD (*Feng and Irvine, 2009*; *Sopko et al., 2009*). Also, the palmitoyltransferase approximated (App) is needed for D localization to the membrane (*Matakatsu and Blair, 2008*). However, for each of these proteins, their role in mediating the regulation of D levels or asymmetry by Ft is not well understood.

Here we describe the *Drosophila* ortholog of the *Fbxl7* gene, which encodes an F-box protein and is a novel component of the Ft signaling pathway. Inactivation of *Fbxl7* results in increased tissue growth via the Hippo pathway and abnormalities in wing shape and proximodistal patterning of appendages. Fbxl7 localizes preferentially to the proximal edge of cells in the wing pouch where it binds to and co-localizes with Ft. We find a role for Fbxl7 in one of the growth-suppressing signaling pathways downstream of Ft and also demonstrate a role for Fbxl7 in regulating the amount of D at the apical membrane as well as its distribution around the edge of the cell.

## Results

### Fbxl7 functions as a negative regulator of tissue growth and modulates signaling via the Hippo pathway

In two different genetic screens, one for mutations that caused cells to outgrow their neighbors (described in *Tapon et al., 2001*) and another for mutations that enabled cells to promote the elimination of their slower-growing neighbors by cell competition (*Hafezi et al., 2012*), we identified mutant alleles of the *Drosophila Fbxl7* gene (*CG4221*), which encodes a protein with an F-box and 11 leucine-rich repeats (LRRs) (*Figure 1A*, *Figure 1—figure supplement 1A*). Fbxl7 has a conserved human ortholog (FBXL7*)* that shares 49% amino acid identity over the region spanning the F-box and the LRRs. Most proteins with these motifs function as part of an SCF-type ubiquitin ligase, a protein complex which polyubiquitylates substrate proteins and targets them for degradation by the proteasome (*Skaar et al., 2013*). A third allele was identified fortuitously in an unrelated stock. Mutant clones of all three alleles were overrepresented in the adult eye when compared to clones of the parental FRT82B chromosome (*Figure 1B*), suggesting that these *Fbxl7* mutations cause increased tissue growth (*Figure 1C–E*). Two of the mutations generate premature stop codons upstream of all conserved domains, while the third causes a cysteine-to-tyrosine change in a conserved residue in one of the LRRs (*Figure 1A*, *Figure 1—figure supplement 1B–C*) that likely interferes with the normal function of the protein, indicating that all three alleles reduce or eliminate *Fbxl7* function. We also found that a *Mi{MIC}* minos insertion in the first intron of *Fbxl7* (*Venken et al., 2011*) results in a strong loss-of-function phenotype similar to our other mutant alleles (*Figure 1—figure supplement 1A*, *Figure 1—figure supplement 2A–B*).

Although clones of mutant cells display a clear growth advantage, flies homozygous for each of these *Fbxl7* mutations are viable and fertile. However, the wings of *Fbxl7* homozygotes or hemizygotes (*Fbxl7*⁻/Deficiency) are larger and more rounded than wild-type wings (*Figure 1F–H*; quantified in *Figure 1L* and *Figure 1—figure supplement 2A*) and the distance between the cross veins is reduced (*Figure 1F–G*, *Figure 1—figure supplement 2B*). The same alterations in wing area and spacing between the cross veins were also observed when *Fbxl7* function was reduced by RNAi (*Figure 1L*, *Figure 1—figure supplement 2A–B*) (*Dui et al., 2012*). The combination of overgrowth and reduced spacing of the cross veins is especially reminiscent of mutations in the Ft branch of the Hippo signaling pathway (*Bryant et al., 1988*; *Mahoney et al., 1991*; *Clark et al., 1995*; *Villano and Katz, 1995*; *Mao et al., 2006*; *Matakatsu and Blair, 2008*; *Mao et al., 2009*).

Since we identified one of the *Fbxl7* alleles in a screen for mutations that made cells capable of eliminating their neighbors (*Hafezi et al., 2012*), we examined imaginal discs for evidence of cell death. We observed elevated levels of activated caspase-3, a marker of apoptosis, especially in wild-type cells adjacent to *Fbxl7* mutant clones (*Figure 1M–M″*). Thus *Fbxl7* mutant cells do indeed behave as supercompetitors similar to loss-of-function mutations in *ft* or in core components of the Hippo pathway such as *hpo* or *wts* (*Tyler et al., 2007*).

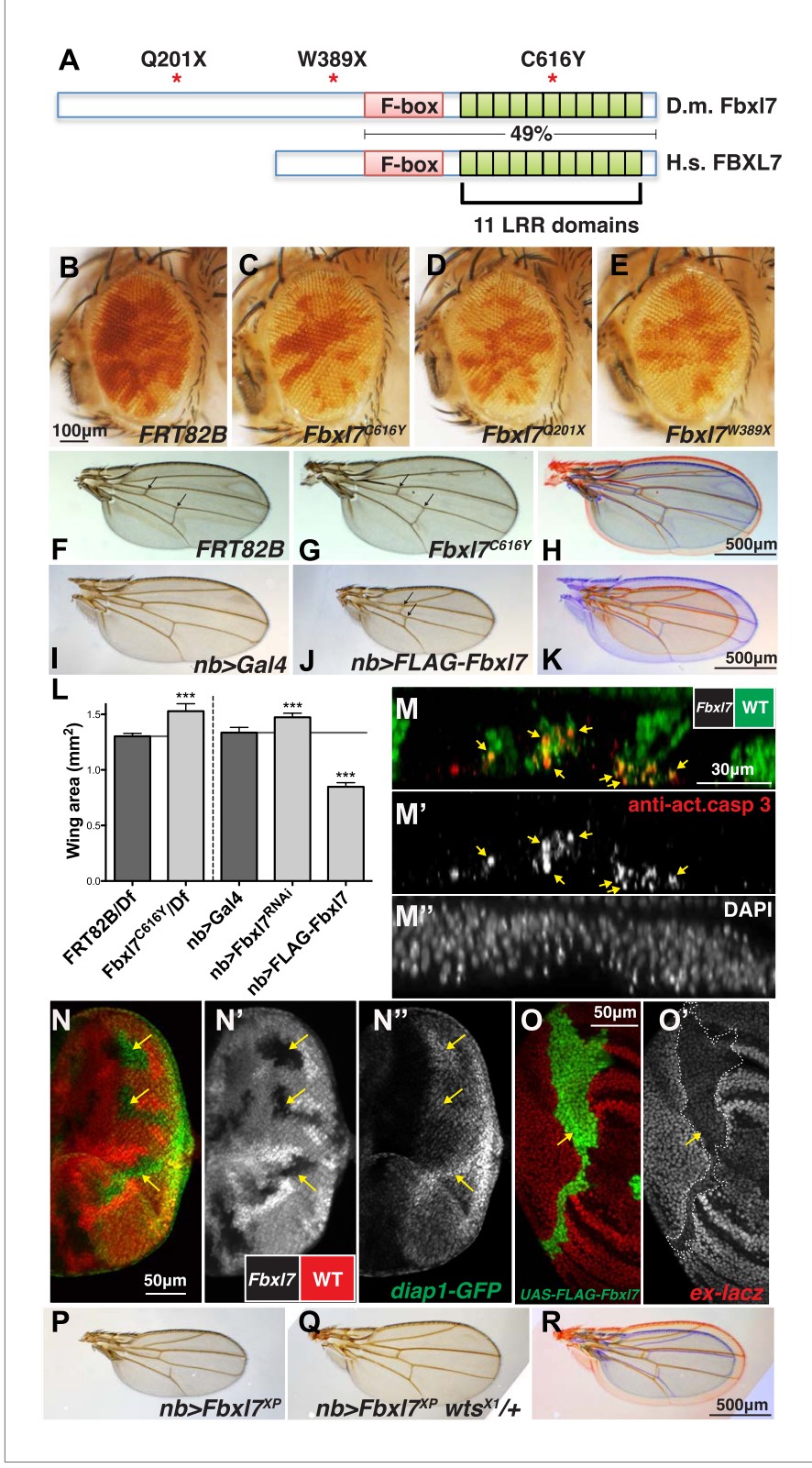

**Figure 1**. Fbxl7 negatively regulates growth through the Hippo pathway. (**A**) Protein model of Drosophila Fbxl7 and Human FBXL7 showing the three alleles identified (red asterisks), F-box, and 11 Leucine Rich Repeat (LRR) domains. The two proteins have 49% amino acid identity throughout the F-box and LRR domains. (**B**–**E**) Mosaic adult eye

*Figure 1. Continued on next page*

*Figure 1. Continued*

assay. Heterozygous and wild-type cells have red pigment and homozygous mutant cells lack pigment. (**B**) Control mosaic eye. (**C**) *Fbxl7^C616Y*, (**D**), *Fbxl7^Q201X* and (**E**) *Fbxl7^W389X* mosaic eyes are composed of more mutant cells. (**F–K**) Adult wings with overlays. Arrows indicate anterior and posterior crossveins. Compared to (**F**) *FRT82B* control wings, (**G**) *Fbxl7^C616Y* homozygous wings are larger and crossveins are closer. (**H**) Merge shows **F** in blue and **G** in red. Compared to (**I**) *nubbin-Gal4 (nb-Gal4)* control wings, (**J**) *nb>FLAG-Fbxl7* overexpressing wings are smaller and crossveins are closer. (**K**) Merge shows **I** in blue and **J** in red. (**L**) Quantification of wing area from *Fbxl7* loss-of-function, RNAi (*JF01515*), and overexpression. n ≥ 20 wings, ***p ≤ 0.001, error bars show SD. (**M–M″**) Cell competition assay in the mosaic eye imaginal disc. (**M**) Wild-type cells are marked by GFP (green), while *Fbxl7* mutant cells are GFP negative. (**M′**) Activated caspase-3 (red) is detected in dying cells that are GFP positive (arrows). (**M″**) DAPI shows all nuclei. (**N–N″**) Mosaic eye imaginal disc with *diap1-GFP* (green) reporter. (**N–N′**) Wild-type cells are marked with RFP (red) and *Fbxl7* mutant cells are RFP negative. (**N″**) Mutant clones show higher levels of *diap1-GFP* (arrows). (**O–O′**) Mosaic wing imaginal disc with *ex-lacZ* reporter (red). A clone overexpressing FLAG-Fbxl7 (green, cells marked by EGFP) has lower levels of *ex-lacZ* (arrow). (**P–R**) Wing size genetic interaction assay. Compared to (**P**) *nb>Fbxl7^XP* alone, (**Q**) reducing the dosage of *wts* partially rescues the small wing phenotype. (**R**) Merge shows **P** blue and **Q** in red.

The following figure supplements are available for figure 1:

**Figure supplement 1**. Fbxl7 gene and protein models.

**Figure supplement 2**. Additional Fbxl7 mutant phenotypes.

Reduced signaling via the Hippo pathway results in increased activity of the transcriptional co-activator Yki. In *Fbxl7* mutant clones in the eye imaginal disc, expression of a *diap1-GFP* reporter gene (***Zhang et al., 2008***) was increased, especially posterior to the morphogenetic furrow (***Figure 1N–N″***) consistent with increased Yki activity. Additionally, the enlarged wing phenotype observed upon expression of *Fbxl7^RNAi* was enhanced by heterozygosity of the *wts^X1* allele (***Figure 1—figure supplement 2K–L***). Together these results indicate that loss of *Fbxl7* leads to increased growth via the Hippo pathway.

When we overexpressed *Fbxl7* in the wing imaginal disc, the adult wings were smaller and had a reduced distance between the cross veins (***Figure 1I–L***, ***Figure 1—figure supplement 2A–B***). Over-expressing Fbxl7 also reduced the length of distal leg segments (***Figure 1—figure supplement 2C–D***). Results were similar using either a *UAS-Fbxl7* transgene or a *P[XP]* transposon which contains UAS sequences upstream of the endogenous *Fbxl7* transcriptional start site (***Figure 1—figure supplement 1A***). This reduction in wing size was suppressed by heterozygosity of the *wts^X1* allele (***Figure 1P–R***, ***Figure 1—figure supplement 2A***). When we overexpressed a form of *Fbxl7* bearing the missense mutation identified in the screen, *Fbxl7^C616Y*, there was, if at all, a slight increase in wing size (***Figure 1—figure supplement 2A***) suggesting that this mutation disrupts the normal function of the protein and likely functions as a dominant-negative mutation at least under conditions of overexpression.

Since increased signaling via the Hippo pathway would reduce Yki activity, we examined expression of Yki reporters. Overexpression of *Fbxl7* reduced expression of an *ex-lacZ* reporter (***Boedigheimer and Laughon, 1993***; ***Hamaratoglu et al., 2006***) in a cell-autonomous manner (***Figure 1O–O′***). However, wild-type cells close to the *Fbxl7*-overexpressing cells had increased *ex-lacZ* reporter expression, especially in the notum of the wing disc and in the eye disc (***Figure 1—figure supplement 2M–N′***). A similar phenomenon was observed with the *bantam* sensor (***Figure 1—figure supplement 2O–O′***) (***Brennecke et al., 2003***), which is expressed at higher levels when Yki activity is reduced. This non-autonomous increase in Yki activity is similar to that seen when Ft is overexpressed (***Matakatsu and Blair, 2012***) or at boundaries of differential Ds or Fj activity (***Willecke et al., 2008***). Taken together, these results indicate that Fbxl7 functions as a negative regulator of growth via the Hippo pathway. Moreover, the multiple phenotypic similarities between alterations in Ft levels and Fbxl7 levels suggest that Fbxl7 functions in proximity to Ft.

## Fbxl7 localizes to the apical membrane and is distributed asymmetrically

A polyclonal antibody to an N-terminal portion of Fbxl7 detects uniform Fbxl7 expression throughout the wing imaginal disc (***Figure 2A***), with a slight enrichment at the dorsal-ventral boundary in the pouch as is also observed for Ft protein (***Mao et al., 2009***). At the cellular level, punctate staining is

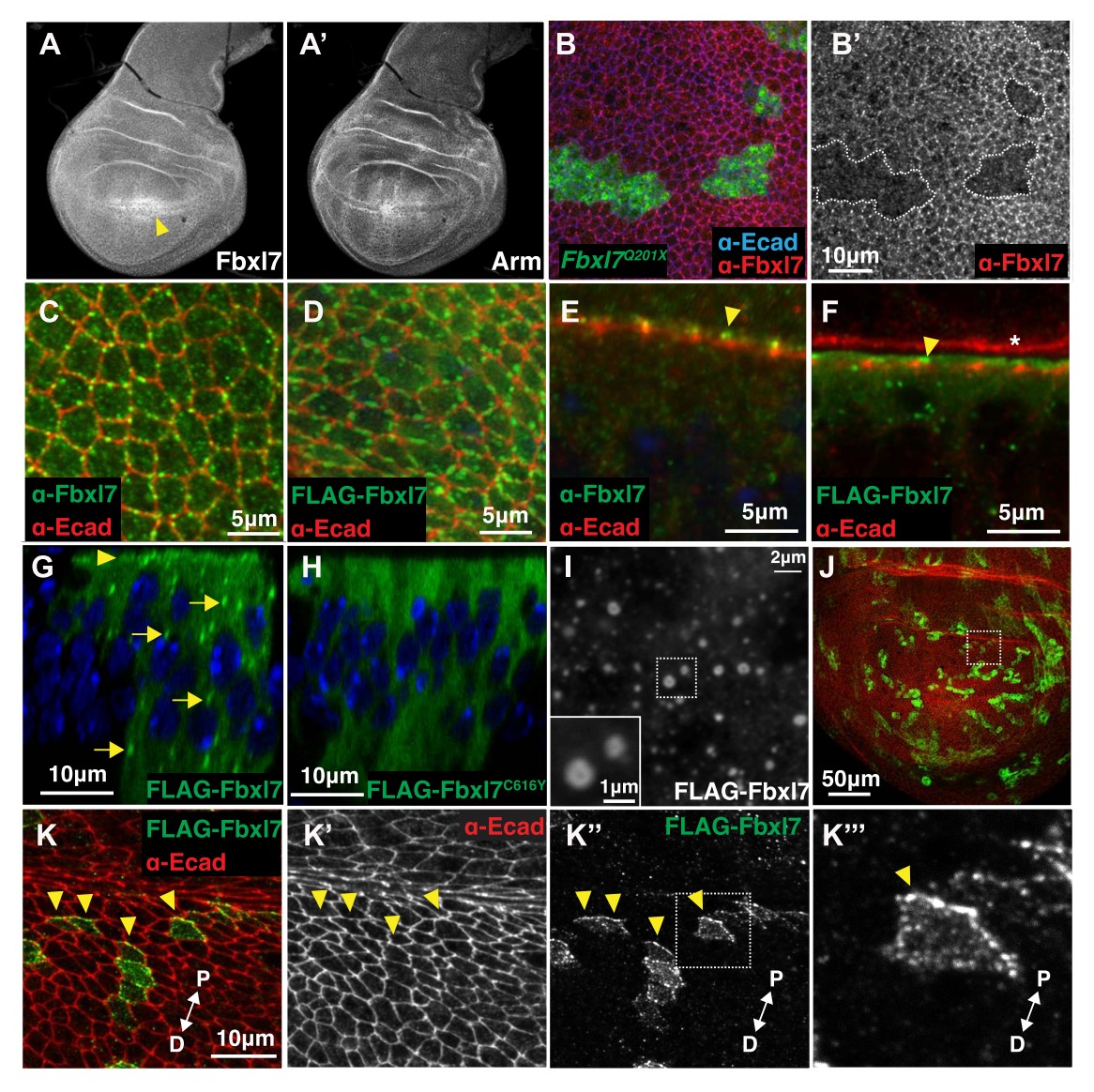

**Figure 2**. Fbxl7 is localized to apical membrane, cytoplasmic puncta, and the proximal side of planar polarized cells. Confocal slice of (**A**) endogenous Fbxl7 and (**A'**) Armadillo (Arm) in the wing imaginal disc. Arrow indicates enrichment of Fbxl7 at the dorso-ventral boundary. (**B–B'**) A confocal slice through the apical surface of wing disc cells. Fbxl7 (red) accumulates at the apical membrane and is lost from MARCM *Fbxl7^{Q201X}* clones (green). (**C–F**) Endogenous Fbxl7 and expressed FLAG-Fbxl7 (green) are localized to apical puncta aligned with cell edges marked by E-cadherin (E-cad) (red). (**C–D**) Confocal slices through the apical surface of wing disc cells. (**E–F**) Confocal slice through folds in the wing disc. Fbxl7 is apical to E-cad (arrowheads). (**F**) Asterisk indicates adjacent fold that does not express FLAG-Fbxl7. (**G–H**) Confocal Z-slice through the wing disc with clones of cells expressing FLAG-Fbxl7 or FLAG-Fbxl7^{C616Y} (green). Nuclei are shown with DAPI (blue). (**G**) FLAG-Fbxl7 localizes to apical membrane (arrowhead) and cytoplasmic puncta (arrows), whereas (**H**) FLAG-Fbxl7^{C616Y} shows diffuse cytoplasmic localization. (**I**) Confocal section through peripodial membrane showing FLAG-Fbxl7 localization to hollow puncta. Inset shows higher magnification of outlined box. (**J–K'''**) Confocal slice of the wing disc pouch stained for E-cad (red) with clones expressing FLAG-Fbxl7 (green). (**K–K''**) Magnified region from box in **J**, showing FLAG-Fbxl7 enriched on proximal membrane (arrowheads). (**K''''**) Magnified region from box in **K'''**. D = distal, P = proximal.

observed outlining the apical profiles of cells, which is absent in homozygous mutant clones of the *Fbxl7^{Q201X}* allele (**Figure 2B–B'**) indicating that the truncated protein generated by this allele is likely unstable. In *Fbxl7^{C616Y}* clones, apical puncta are absent but cytoplasmic staining is observed above background levels, indicating that the mutant protein is present but does not localize apically (not shown). An Fbxl7 protein with an N-terminal FLAG epitope tag (FLAG-Fbxl7) exhibits an apical

localization that is very similar to that of the endogenous protein (*Figure 2C–F*). Using either the anti-Fbxl7 antibody (*Figure 2C,E*) or FLAG-Fbxl7 (*Figure 2D,F*), we found that Fbxl7 localizes to the sub-apical region of cells, apical to the adherens junctions marked by E-cadherin. FLAG-Fbxl7 is also found in intracellular puncta (*Figure 2G*). In contrast, FLAG-Fbxl7 protein bearing the C616Y missense muta-tion displays only diffuse cytoplasmic localization (*Figure 2H*) suggesting that the normal function of Fbxl7 may be contingent upon its localization to the apical region or cytoplasmic puncta. In *Drosophila* S2 cells (not shown) or the flattened cells of the peripodial epithelium (*Figure 2I*), confocal sections show puncta with diameters typically in the range of 400–500 nm (some as large as 1000 nm) with a hollow interior, consistent with the possibility that these might be vesicles.

In cells of the wing imaginal disc, Ft is preferentially expressed on the proximal side of cells and Ds and D on the distal surface (*Ambegaonkar et al., 2012*; *Brittle et al., 2012*). We generated small clones that expressed FLAG-Fbxl7, which enabled us to examine the borders between FLAG-Fbxl7-expressing cells and wild-type cells. In the dorsal part of the wing pouch, where polarization of D is most evident (*Brittle et al., 2012*), FLAG-Fbxl7 localizes preferentially to the proximal side of cells (*Figure 2J,K–K‴*).

## Fbxl7 associates with Ft and regulates its localization

Since the localization of Fbxl7 is similar to that described for Ft, we examined whether the two proteins co-localize. Both anti-Fbxl7 and anti-Ft revealed apical staining in a punctate manner with a considerable degree of overlap (*Figure 3A–A″*). Additionally, we observed co-localization of FLAG-Fbxl7 and Ft at the apical membrane (*Figure 3B–B″*) as well as in cytoplasmic puncta (*Figure 3B″–B‴*, *Figure 3—figure supplement 1A*), many of which were basally located. Higher gain settings were required to visualize the comparatively faint Ft staining in puncta (*Figure 3B‴*). Because of a higher background level of cytoplasmic staining with anti-Fbxl7, the FLAG-tagged Fbxl7 protein was neces-sary to observe co-localization in puncta.

To determine whether Ft and Fbxl7 can interact physically, we co-transfected S2 cells with tagged versions of Fbxl7 and a portion of Ft that includes the transmembrane domain and the entire intracel-lular domain (FatICD). FatICD co-immunoprecipitates with FLAG-Fbxl7, whereas association of FatICD with FLAG-Fbxl7$^{C616Y}$ is greatly reduced (*Figure 3C*). We also examined the ability of truncated Fbxl7 proteins to interact with Ft and find that Fbxl7 interacts with Ft mostly via its LRRs (*Figure 3—figure supplement 1B–C*). A weaker interaction is also observed between Ft and the N-terminal portion of Fbxl7. Thus wild-type Fbxl7 can associate, either directly or indirectly, with the intracellular domain of Ft and this interaction mostly occurs via the LRRs of Fbxl7.

The apical localization of Fbxl7 was absent in *ft* clones (*Figure 3D–D‴*). However, an increase in diffuse cytoplasmic staining was observed (*Figure 3E–E‴*). Thus the localization of Fbxl7 to the apical region is dependent upon Ft and in the absence of Ft, Fbxl7 re-localizes to the cytoplasm. Since Ft and Fbxl7 also co-localize to cytoplasmic puncta or vesicles, we examined whether this localization of Fbxl7 also depends on Ft. Surprisingly, unlike the apical localization, punctate localization of FLAG-Fbxl7 was still observed in *ft* clones indicating that the localization of Fbxl7 in these cytoplasmic puncta is independent of Ft (*Figure 3F–F″*).

Since proteins similar to Fbxl7 often bind to their substrates via their LRRs and promote their polyubiquitylation and degradation (*Skaar et al., 2013*), we tested the effect of changes in Fbxl7 on the levels and localization of Ft. Increasing Fbxl7 levels resulted in clearly increased levels of apical Ft (*Figure 3G–G′*) and slightly increased cytoplasmic staining of Ft (*Figure 3—figure supplement 1A*). Surprisingly, a slight elevation of apical Ft levels was also observed in *Fbxl7* mutant clones (*Figure 3H–H″*). The overall levels of Ft protein in imaginal discs, as assessed by Western blotting, were not obvi-ously changed in either case (*Figure 3—figure supplement 1D*). These results are inconsistent with Fbxl7 promoting Ft degradation and instead suggest that Fbxl7 regulates Ft localization. In sup-port of this, we do not observe an obvious increase in Ft ubiquitylation from expressing Fbxl7 in S2 cells (not shown).

## The apical localization of Fbxl7 does not require Ds or Dco

Since the phenotypic abnormalities of *Fbxl7* mutants resemble those of hypomorphic alleles of *ft*, and the recruitment of Fbxl7 to the apical region of the cell is dependent upon Ft, we explored the rela-tionship between Fbxl7 and proteins known to regulate Ft in more detail. In *ds* mutant clones, apical localization of Fbxl7 is no longer observed as discrete puncta at cell edges but is rather more diffuse

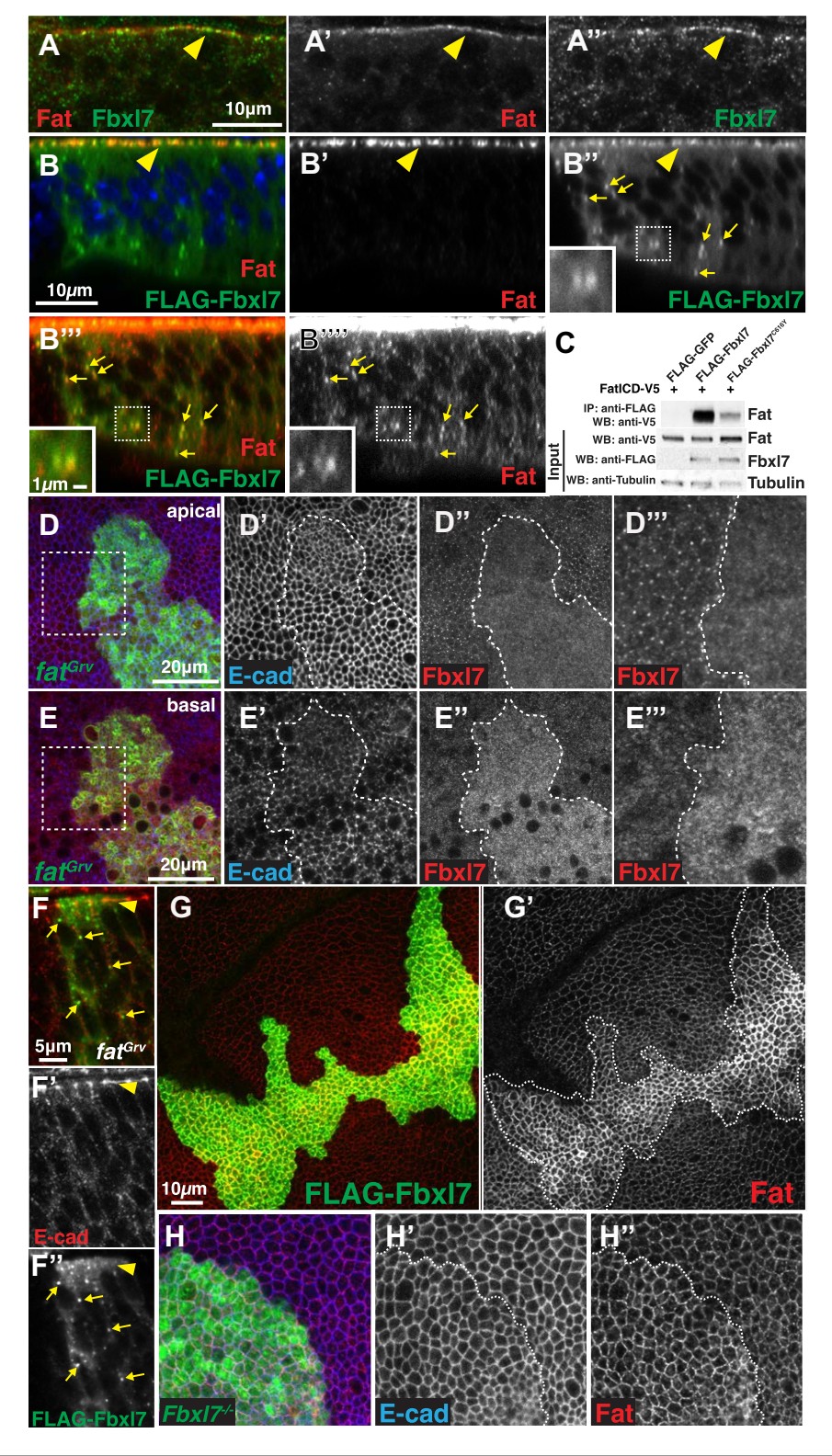

**Figure 3**. Fbxl7 physically interacts with Fat and regulates its apical localization. (**A–A"**) Confocal slice through a wing disc fold showing endogenous Fbxl7 (green) and Fat (red) co-localize at apical membrane (arrowhead). (**B–B""**) Confocal Z-section showing FLAG-Fbxl7 (green) and Fat (red) co-localize at (**B–B"**) apical membrane (arrowhead) and (**B"–B""**) cytoplasmic puncta (arrows). (**B"–B""**) Inset shows magnification of puncta. **B""** uses higher gain settings

*Figure 3. Continued on next page*

*Figure 3. Continued*

than **B**′ to visualize Fat in puncta. (**C**) Co-immunoprecipitation experiment in S2 cells. FatICD-V5 pulls down with FLAG-Fbxl7, whereas pulldown is reduced with FLAG-Fbxl7$^{C616Y}$. (**D–D**‴) Confocal slice of the wing disc at the apical surface. Apical Fbxl7 (red) localization is lost from MARCM *fat*$^{Grv}$ clones (green), whereas (**D**′) E-cad (blue) localization is unchanged. (**D**‴) shows magnification of the box in **D**. (**E–E**‴) A basal confocal slice through the same clone in **D**, showing increased cytoplasmic levels of Fbxl7. (**F–F**″) Confocal slice through a fold showing a MARCM *fat*$^{Grv}$ clone (GFP marker not shown) which expresses FLAG-Fbxl7 (anti-Flag, green). (**F**′) E-cad (red) marks apical membrane. FLAG-Fbxl7 is not apically localized in *fat*$^{Grv}$ clones (arrowhead), but does localize to cytoplasmic puncta (arrows). (**G–G**′) Confocal slice through the apical surface of a disc overexpressing FLAG-Fbxl7 (green) in clones. (**G**′) Apical Fat (red) levels are elevated within the clone. (**H–H**″) Confocal slice through the apical surface with a MARCM *Fbxl7*$^{Q201X}$ clone (green) showing (**H**′) no change in levels of apical E-cad (blue) and (**H**″) slightly elevated levels of apical Fat (red).

The following figure supplement is available for figure 3:

**Figure supplement 1**. Additional analysis of the relationship between Fbxl7 and Fat.

---

(*Figure 4A–A*″). Moreover, in contrast to *ft* clones, we do not see an increase in cytoplasmic Fbxl7 in *ds* clones at more basal focal planes, indicating that Fbxl7 is still predominantly at an apical location (*Figure 4B–B*″). These changes in Fbxl7 localization could simply be a consequence of the more diffuse localization of Ft that is observed in *ds* clones (*Strutt and Strutt, 2002*; *Ma et al., 2003*; *Mao et al., 2009*). Fj is required for normal localization of Ds and Ft (*Strutt and Strutt, 2002*; *Ma et al., 2003*). In agreement with this, we see subtle effects on Fbxl7 localization in *fj* clones, which appears similar to that seen in *ds* clones (*Figure 4—figure supplement 1A*).

When Fbxl7 is overexpressed in clones, cells have more prominent apical expression of Ds in puncta (*Figure 4C–C*″). Additionally, in wild-type cells bordering the Fbxl7-overexpressing clone, Ds staining is reduced and accumulates in prominent puncta at the surface that abuts the Fbxl7-overexpressing cells (*Figure 4D–E*″). Given that Ds can be drawn toward cells with greater levels of Ft (*Ma et al., 2003*), Ds may be drawn toward Fbxl7-overexpressing cells due to the increased Ft levels. Furthermore, the puncta of Ds in adjacent wild-type cells are in register with Fbxl7 puncta, consistent with the coupling of Ds in wild-type cells to Fbxl7-bound Ft within the clone. In *Fbxl7* mutant clones, there is, at best, a very slight elevation of Ds levels (*Figure 4F–F*″). Thus, the effects of Fbxl7 on Ds levels are minor compared to the effects on Ft levels. Additionally, we could not detect Ds in immunoprecipitates of Fbxl7 when the two proteins were co-expressed in S2 cells (not shown). Together, these findings suggest that Fbxl7 binds to and functions with Ft rather than Ds. Despite this, we did observe co-localization of Fbxl7 and Ds at apical membranes and in more basally located cytoplasmic puncta (*Figure 4G–G*″). In the absence of evidence for direct interactions between Fbxl7 and Ds, their co-localization, at least at the cell surface, may result from Fbxl7 bound to Ft that is in turn bound to Ds.

Ds binding to Ft induces the phosphorylation of the ICD of Ft, which requires the protein kinase, Dco (*Feng and Irvine, 2009*; *Sopko et al., 2009*). Since some F-box proteins bind to phosphorylated proteins (*Skaar et al., 2013*), we tested whether the apical localization of Fbxl7 was dependent upon Dco function. The apical localization of Fbxl7 was not obviously changed in clones of the *dco*$^3$ allele that is unable to phosphorylate Ft (*Figure 4H–H*″) (*Sopko et al., 2009*). While Dco is capable of binding to Fbxl7 as assessed by co-immunoprecipitation from S2 cells (*Figure 4J*), the apical localization of Fbxl7 was still observed in clones of the null *dco* allele, *dco*$^{le88}$ (*Figure 4I–I*″), thus indicating that Dco function is altogether unnecessary for the apical localization of Fbxl7.

Furthermore, while changes in Fbxl7 alter Hippo signaling, changes in Hippo signaling do not regulate Fbxl7 levels or localization, as Fbxl7 localization is normal in clones mutant for *dachs* or *wts* (*Figure 4—figure supplement 1B–C*″).

## Fbxl7 functions in one of two growth-suppressing pathways downstream of Ft

The primary amino acid sequence of the ICD of Ft does not predict any domains with enzymatic activity or known protein–protein interaction motifs. Hence, it has not been easy to understand how it functions in signal transmission. However, six blocks of sequence (labeled A–F in *Figure 5A* based on the nomenclature of *Pan et al. (2013)*) are conserved with the ICD of mammalian Fat4. A region between

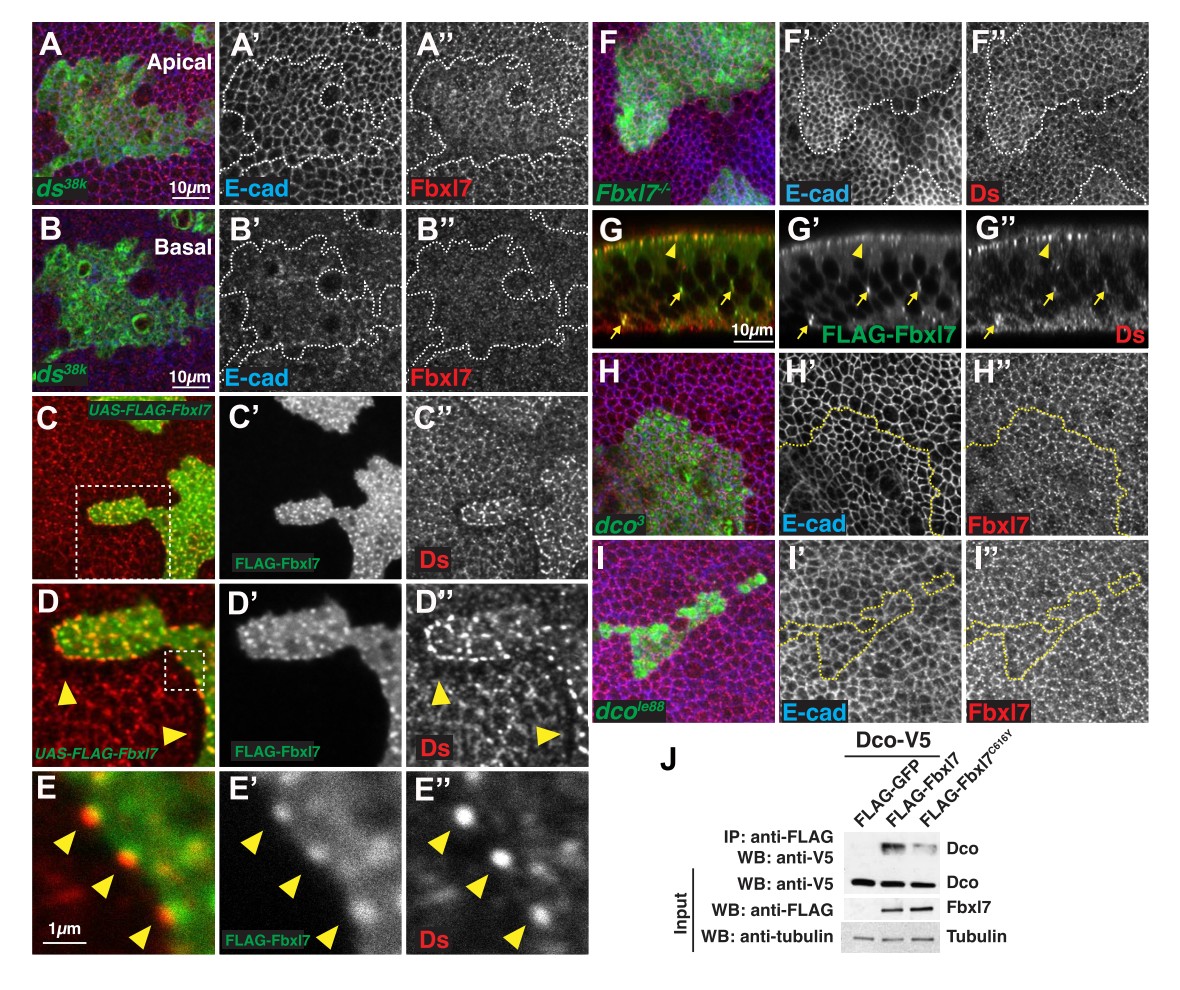

**Figure 4**. Relationship between Fbxl7 and the Fat pathway proteins Ds and Dco. (**A–A″**) Confocal slice through the apical surface of a disc with MARCM *ds*[38k] clones (green) showing disturbed localization of Fbxl7 (red). E-cad staining is not altered (blue) (**B–B″**) A basal confocal slice through the same clone in **A**, showing no change in Fbxl7 cytoplasmic levels. (**C–E″**) Confocal slice through the apical surface of a disc with FLAG-Fbxl7 overexpressing clones (green) and stained for Ds (red). (**C–D″**) Apical Ds levels appear higher and more punctate in FLAG-Fbxl7 expressing clones. Wild-type cells immediately adjacent to the clone have reduced apical Ds (arrowheads). (**E–E″**) Ds and FLAG-Fbxl7 puncta are aligned on either side of the clone boundary (arrowheads). (**F–F″**) Apical confocal slice of a disc containing MARCM *Fbxl7*[Q201X] clones (green) and stained for Ds (red) and E-cad (blue). Ds levels are normal or slightly elevated, in clones. (**G–G″**) Confocal Z-section of a clone expressing FLAG-Fbxl7 (green) and stained for Ds (red). Both are localized to apical membrane (arrowhead) and frequently co-localize in cytoplasmic puncta (arrows). (**H–I″**) Apical confocal slice of MARCM *dco*[3] or *dco*[le88] clones (green) and staining for Fbxl7 (red) and E-cad (blue). Apical Fbxl7 levels are unchanged in (**H–H″**) *dco*[3] and (**I–I″**) *dco*[le88] clones. (**J**) Co-immunoprecipitation experiment in S2 cells. Dco-V5 pulls down with FLAG-Fbxl7, whereas pulldown is reduced with FLAG-Fbxl7[C616Y].

The following figure supplement is available for figure 4:

**Figure supplement 1**. Additional analysis of the relationship between Fbxl7 and Fat pathway proteins.

the conserved blocks 'B' and 'C' seems necessary for the major growth-suppressive function of Ft (*Matakatsu and Blair, 2012*; *Bossuyt et al., 2013*; *Zhao et al., 2013*). In our screen, we identified an allele of *ft*, *ft*[61] (*Figure 5A*), which displays strong overgrowth (*Figure 5C,K*) and is caused by a single amino acid change (T to I) within this region. *ft*[61] displays phenotypic abnormalities that are very similar to those described for *ft*[sum], which also changes a single amino acid two residues N-terminal to *ft*[61] (*Bossuyt et al., 2013*). Additionally, in a *ft* null background, deletion of one of the conserved blocks (block D in *Figure 5A*) in a *ft* genomic rescue transgene was shown to cause overgrowth (*Pan et al., 2013*) albeit to a much lesser extent than for *ft*[61] and *ft*[sum]; flies had slightly overgrown, rounder wings with decreased spacing between the crossveins (*Figure 5E*).

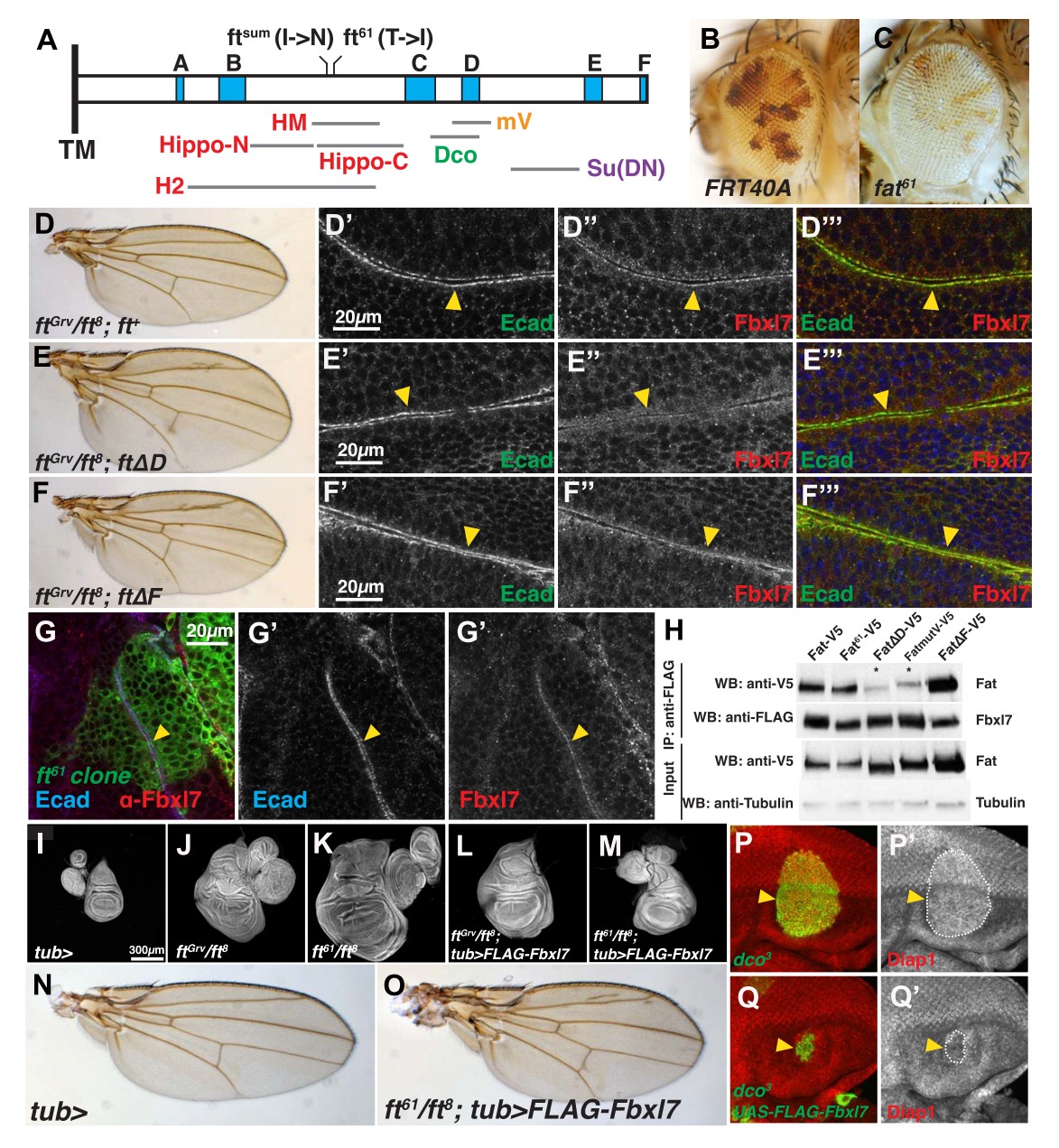

**Figure 5**. Fbxl7 functions in one of two growth-suppressing pathways downstream of Ft. (**A**) Protein model of the intracellular domain of Fat showing the transmembrane domain (TM), regions conserved with mammalian Fat4 (blue, A–F) (defined by *Pan et al., 2013*), regions associated with the major growth suppressive function of Fat (red) (HM, *Bossuyt et al., 2013*; Hippo-N, Hippo-C, *Matakatsu and Blair, 2012*; H2, *Zhao et al., 2013*), region required for Dco binding (green) (*Sopko et al., 2009*), mutV region (orange) (*Pan et al., 2013*), Su(DN) region (purple) (*Matakatsu and Blair, 2012*), and two point mutations, *ft*$^{sum}$ (*Bossuyt et al., 2013*) and *ft*$^{61}$ (this study). Size and position of regions are drawn to scale relative to the ICD. (**B–C**) Mosaic adult eye assay. Heterozygous wild-type cells have red pigment and homozygous mutant cells lack pigment. Compared to (**B**) control *FRT40A* mosaic eyes, (**C**) *ft*$^{61}$ mosaic eyes are larger and have more mutant tissue. (**D**) *ft*$^{Grv}$/*ft*$^{8}$; *ft*$^{+}$ adult wing and (**D'–D'''**) confocal slice of a wing disc showing that Fbxl7 (red) is localized to the apical membrane similar to E-cad (green). (**E**) *ft*$^{Grv}$/*ft*$^{8}$; *ft*Δ*D* adult wing and (**E'–E'''**) confocal slice showing that Fbxl7 (red) apical localization is disrupted. (**F**) *ft*$^{Grv}$/*ft*$^{8}$; *ft*Δ*F* adult wing and (**F'–F'''**) confocal slice showing that Fbxl7 (red) apical localization is normal and similar to that in **D'–D'''**. (**G**) Confocal slice of a disc containing a MARCM *ft*$^{61}$ clone (green) and stained for Fbxl7 (red) and E-cad (blue). Fbxl7 apical localization is normal in *ft*$^{61}$ cells (**H**) Co-immunoprecipitation experiment in S2 cells. Fat-V5, Fat$^{61}$-V5, and FatΔF-V5 pull down with FLAG-Fbxl7, whereas pull down of FatΔD-V5 and FatmutV-V5 is reduced. Expressed Fat proteins contain only transmembrane and cytoplasmic regions (ICD). (**I–M**) Wing imaginal discs (and associated leg and haltere discs) at low magnification. Compared to (**I**) control *tub-Gal4* discs, (**J**) *ft*$^{Grv}$/*ft*$^{8}$ and (**K**) *ft*$^{61}$/*ft*$^{8}$ discs are larger and have more folds. (**L**) Ubiquitous expression of Fbxl7 does not rescue *ft*$^{Grv}$/*ft*$^{8}$ disc overgrowth. (**M**) Ubiquitous expression of Fbxl7 rescues disc overgrowth

*Figure 5. Continued on next page*

*Figure 5. Continued*

of *ft⁶¹/ft⁸*. (**N–O**) Adult wing from (**N**) control *tub-Gal4* and (**O**) ubiquitous expression of FLAG-Fbxl7 in an *ft⁶¹/ft⁸* background. Animal lethality is rescued. (**P–Q'**) Confocal slice through the eye imaginal disc showing MARCM clones (green) and anti-Diap1 staining (red). (**P–P'**) *dco³* clones have elevated Diap1 levels and are overgrown, whereas (**Q–Q'**) *dco³* clones expressing FLAG-Fbxl7 have wild-type Diap1 levels and are reduced in size.

The following figure supplements are available for figure 5:

**Figure supplement 1**. Additional images of Fbxl7 localization in Fat deletion backgrounds.

**Figure supplement 2**. Domain D of Ft is required for the effects of Fbxl7 on Ft localization.

**Figure supplement 3**. Additional characterization of Fbxl7 rescue experiments.

In contrast to null alleles of *ft,* which display strong overgrowth and cause lethality well before the adult stage, flies lacking *Fbxl7* function are viable and fertile but have slightly overgrown wings that are rounded and have decreased spacing between the cross veins. Thus, their phenotypic abnormalities are very similar to those observed when the *ft* D region is deleted (*ftΔD*). We therefore examined the localization of Fbxl7 in a *ftΔD* background. When a heteroallelic combination of null *ft* alleles, *ftᴳʳᵛ/ft⁸*, is rescued by a wild-type version of *ft (ft⁺)*, wings are normal (*Figure 5D*) and Fbxl7 displays normal apical localization (*Figure 5D'–D'''*, *Figure 5—figure supplement 1*). However, apical localization of Fbxl7 is markedly reduced in *ftᴳʳᵛ/ft⁸; ftΔD* (*Figure 5E'–E'''*). We also examined a different deletion, *ftΔF*, in which wings from these flies are not enlarged but have greatly reduced spacing between the cross veins (*Figure 5F*). In *ftᴳʳᵛ/ft⁸; ftΔF* imaginal discs, the apical localization of Fbxl7 is not disrupted (*Figure 5F'–F'''*). Similarly in *ft⁶¹* clones, which display strong overgrowth, Fbxl7 localization was normal (*Figure 5G–G''*). Thus, the apical localization of Fbxl7 requires the Ft D domain but neither the F domain nor the motif that is disrupted by the *ft⁶¹* allele.

To examine whether the effects on Fbxl7 localization in vivo correlated with the ability of Fbxl7 to physically interact with Ft, we tested the ability of these mutant Ft proteins to co-immunoprecipitate with FLAG-Fbxl7 (*Figure 5H*). Indeed, Ft⁶¹ and FtΔF proteins co-immunoprecipitated at levels comparable to wild-type Ft. However, the level of FtΔD in FLAG-Fbxl7 immunoprecipitates was greatly reduced, as was that of Ftᵐᵘᵗⱽ, a mutant Ft protein in which a cluster of 10 serine/threonine residues overlapping the D domain was mutated to alanines. These sites were identified as candidates for phosphorylation by Dco (*Pan et al., 2013*). However, since Fbxl7 localizes normally in *dco* mutant clones, the inability of Fbxl7 to bind to Ftᵐᵘᵗⱽ might be caused by a change in its conformation that does not depend on phosphorylation by Dco. Indeed *ftᴳʳᵛ/ft⁸; ftᵐᵘᵗⱽ* flies also have phenotypic abnormalities that are very similar to those of *Fbxl7* mutants (*Pan et al., 2013*).

To test for a functional relationship between the D domain of Fat and Fbxl7, we monitored apical levels of Ft, FtΔD, and FtΔF under conditions of Fbxl7 overexpression. Ft and FtΔF levels are increased in cells overexpressing Fbxl7, while FtΔD levels do not increase (*Figure 5—figure supplement 2*). This demonstrates that the D domain is required for Fbxl7 to physically interact with and exert its effects on Ft localization.

If Ft⁶¹ protein is still capable of recruiting Fbxl7 to its apical location, then overexpression of Fbxl7 might suppress the overgrowth observed in mutant discs. The overgrowth and lethality of a *ft* null background (*ftᴳʳᵛ/ft⁸*) can be rescued by ubiquitous expression of Ft (*Matakatsu and Blair, 2012*; *Bossuyt et al., 2013*; *Pan et al., 2013*; *Zhao et al., 2013*; *Figure 5—figure supplement 3D,I*). While ubiquitous Fbxl7 expression was unable to suppress *ftᴳʳᵛ/ft⁸* phenotypes (*Figure 5L*), the overgrowth and lethality of *ft⁶¹/ft⁸* discs was indeed suppressed, resulting in viable adult flies (*Figure 5M,O*). *dco³* and *wts* mutant cells in the eye imaginal disc are overgrown and express higher Diap1 levels, an indicator of Yki activity (*Figure 5P–P'*, *Figure 5—figure supplement 3L–N'*). Fbxl7 overexpression can rescue both clone size and Diap1 levels in *dco³* mutant cells (*Figure 5Q–Q'*, *Figure 5—figure supplement 1O–O'*), but not *wts* clones (*Figure 5—figure supplement 3M–M'*). Thus, mutant Ft⁶¹ protein, or Ft protein that cannot be phosphorylated by Dco, can still bind to Fbxl7 and facilitate the growth-suppressive functions of Fbxl7. Taken together, these findings implicate Fbxl7 in one of two growth-suppressive pathways downstream of Ft and suggest that these two pathways might converge further downstream ('Discussion').

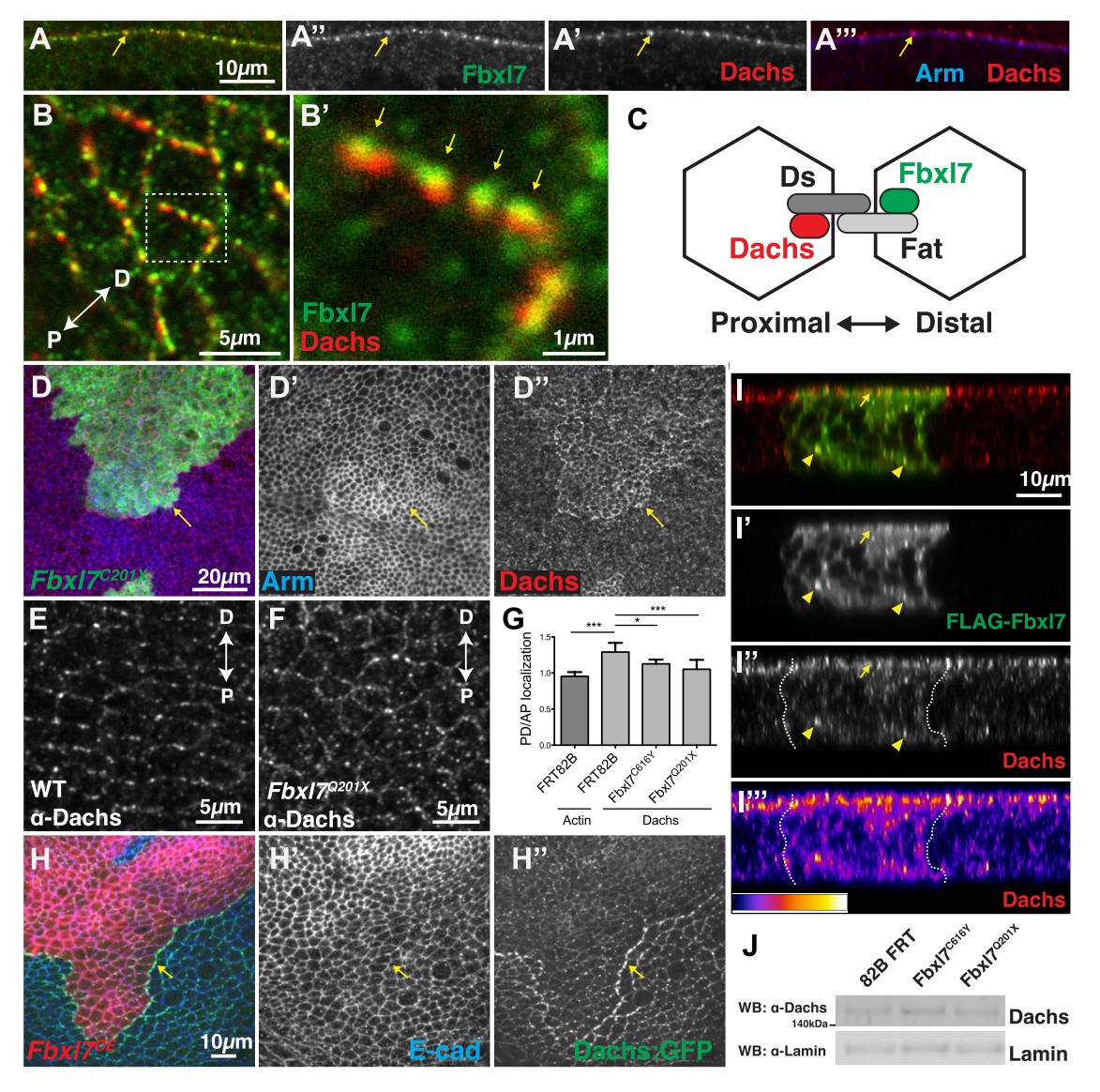

**Figure 6**. Fbxl7 regulates the localization of Dachs. (**A–A"**) Confocal slice through a bend in the wing disc showing (**A'**) Dachs (red) and (**A"**) Fbxl7 (green) localize at subapical membrane. (**A'''**) Like Fbxl7, Dachs is apical to the adherens junction marked by Arm (blue). (**B–B'**) Confocal slice through the apical surface of the wing disc, specifically the dorsal edge of the pouch, showing Dachs (red) and Fbxl7 (green) staining. Dachs and Fbxl7 puncta abut each other on either side of the cell boundary. Proximodistal axis indicated as P<−>D. (**C**) Diagram of polarized wing disc cells in which Dachs is enriched on the distal side and Fbxl7 is on the proximal side, linked by their association to Dachsous and Fat, respectively, which bind across cells. (**D–D"**) Apical confocal slice of MARCM *Fbxl7Q201X* clones (green) and staining for Dachs (red) and Arm (blue). Dachs levels are elevated in clones. (**E–G**) Apical confocal slice with staining for Dachs in (**E**) wild-type or (**F**) *Fbxl7Q201X* discs. Images are from the dorsal edge of the pouch and are aligned so the proximodistal axis is vertical. Dachs enrichment on P/D membrane, seen in (**E**) wild-type discs, is impaired in (**F**) *Fbxl7Q201X* discs. (**G**) Quantification of Dachs P/D enrichment in wing discs. Dachs is localized in a P/D direction, whereas Actin is not. Dachs P/D asymmetry is impaired in both *Fbxl7C616Y* and *Fbxl7Q201X* discs. Significance calculated with one-way ANOVA test. ***p ≤ 0.001, *p ≤ 0.05. Error bars indicate SD. (**H–H"**) Apical confocal slice of FLAG-Fbxl7 overexpressing clones (red, cells marked by RFP) and staining for anti-GFP (green, Dachs:GFP) and E-cad (blue). Apical Dachs levels within the clone are reduced, and Dachs is enriched at the edge of the clone. (**I–I'''**) Confocal z-section of a wing disc with a FLAG-Fbxl7 expressing clone (green) and stained for Dachs (red). FLAG-Fbxl7 and Dachs co-localize to apical membrane (arrow) and intracellular puncta (arrowheads). (**I"–I'''**) Cytoplasmic levels of Dachs are slightly elevated within the clone. (**I'''**) Heat map of **I"**. (**J**) Western blots from wing disc lysates. Endogenous Dachs protein levels are not changed in *Fbxl7* mutant wing discs compared to control.

The following figure supplement is available for figure 6:

**Figure supplement 1**. Additional Dachs tissue staining and Dachs levels in wing discs and S2 cells.

## Fbxl7 regulates the localization of the atypical myosin Dachs

Since Ft and Fbxl7 localized preferentially to the proximal side of cells, we compared the localization of Fbxl7 with that of D. In confocal z-sections, D and Fbxl7 co-localize at the subapical membrane in puncta, apical to the adherens junction marker Armadillo (Arm) (*Figure 6A–A′′′*). However, careful examination of these puncta in x-y sections shows that the Fbxl7 and D puncta are slightly offset in the proximodistal direction (*Figure 6B–B′*). D is localized at higher levels at the distal edge of the cell (*Mao et al., 2006*; *Brittle et al., 2012*) where it is likely stabilized by physical interaction with the cadherin Ds (*Bosveld et al., 2012*). Therefore, a likely explanation is that the formation of multimeric Ft–Ds complexes between cells results in the concomitant accumulation of Fbxl7 at the FatICD and D at the DsICD (*Figure 6C*).

To investigate whether Fbxl7 can regulate the levels or localization of D, we first examined *Fbxl7* mutant clones. The levels of apical D are increased throughout the clone (*Figure 6D–D′′*) although not to the extent that occurs in *ft* clones. Thus Fbxl7 negatively regulates the level of D at the apical membrane. To determine whether Fbxl7 has a role in generating or maintaining the asymmetrical distribution of D, we examined the distribution of D in *Fbxl7* mutant wing discs. In these experiments, the distal edge of one cell cannot be distinguished from the proximal edge of its neighbor. However, in wild-type cells, endogenous D is preferentially observed on the proximal/distal edges and is found at lower levels at the other edges (*Brittle et al., 2012*; *Figure 6E,G*). In *Fbxl7^{C201X}* and *Fbxl7^{C616Y}* homozygotes, this bias in the distribution of D within the cells is reduced (*Figure 6F–G*), indicating that Fbxl7 also has a role in regulating the asymmetric localization of D.

We examined the localization of Dachs-GFP in clones that overexpressed *Fbxl7*. In these clones there was reduction in the overall levels of apical D (*Figure 6H–H′′*). In addition, Dachs-GFP puncta (*Figure 6H–H′′*) or endogenous D (*Figure 6—figure supplement 1A*) in neighboring wild-type cells are enriched against the border with *Fbxl7* overexpressing cells, and are aligned with puncta containing FLAG-Fbxl7, reminiscent of Ds staining in *Figure 4E–E′′*. This likely resulted from the elevated levels of Ft in *Fbxl7*-overexpressing clones, which would cause an enrichment of Ds (and hence D) on the surface of wild-type cells contacting the clone. In z-sections, we observed subtle changes in the localization of D within the clone itself (*Figure 6I–I′′′*). There was a slight increase in D throughout the cell, possibly at the expense of some of the bright puncta that are normally observed at the apical region. Furthermore, overexpressing Fbxl7 can rescue the higher apical Dachs levels seen in *ft^{61}* clones (*Figure 6—figure supplement 1B–C*). Thus, overexpression of *Fbxl7* may cause a shift in the overall distribution of D from the apical region to the interior of the cell.

## Changing Fbxl7 levels does not alter the levels of Dachs ubiquitylation

To determine whether Fbxl7 functions as part of an SCF-type ubiquitin ligase, we first tested whether Fbxl7 was capable of interacting with either SkpA or Cul1. In co-transfection experiments in S2 cells, robust interactions were observed in both cases indicating that Fbxl7 likely functions as part of an SCF complex (*Figure 7A*). Furthermore, when Fbxl7 was cotransfected with HA-tagged ubiquitin, and ubiquitylated proteins immunoprecipitated with anti-HA, a high molecular weight smear above the size of wild-type Fbxl7 was observed indicating that Fbxl7 is ubiquitylated under these conditions (*Figure 7B*). This is expected, as F-box proteins that function in SCF complexes are often themselves ubiquitylated (*Galan and Peter, 1999*; *Yen and Elledge, 2008*). Interestingly, Fbxl7^{C616Y}, which is incapable of binding to Ft, is also ubiquitylated suggesting that the incorporation of Fbxl7 into an active SCF complex does not require Ft.

Since Fbxl7 may function as a component of an E3 ubiquitin ligase, the most parsimonious explanation of its function would be that Fbxl7 ubiquitylates Dachs directly and promotes its degradation by the proteasome. However, the overall levels of D are unchanged in *Fbxl7* mutant discs (*Figure 6J*), discs that overexpress Fbxl7, or *ft* mutant discs as assessed by Western blotting (*Figure 6—figure supplement 1D*). In addition, increasing doses of transfected FLAG-Fbxl7 in S2 cells does not affect total levels of Dachs-V5 (*Figure 6—figure supplement 1E*). If at all, a slight increase in Dachs-V5 levels was observed. Since Fbxl7 is localized apically and preferentially localizes to the proximal edge of the cell, Fbxl7 could promote D degradation locally and this may not be reflected in the overall levels of D.

We therefore tested whether Fbxl7 was capable of promoting D ubiquitylation. These experiments were conducted in both S2 cells and imaginal discs. Ubiquitylated D was readily detected.

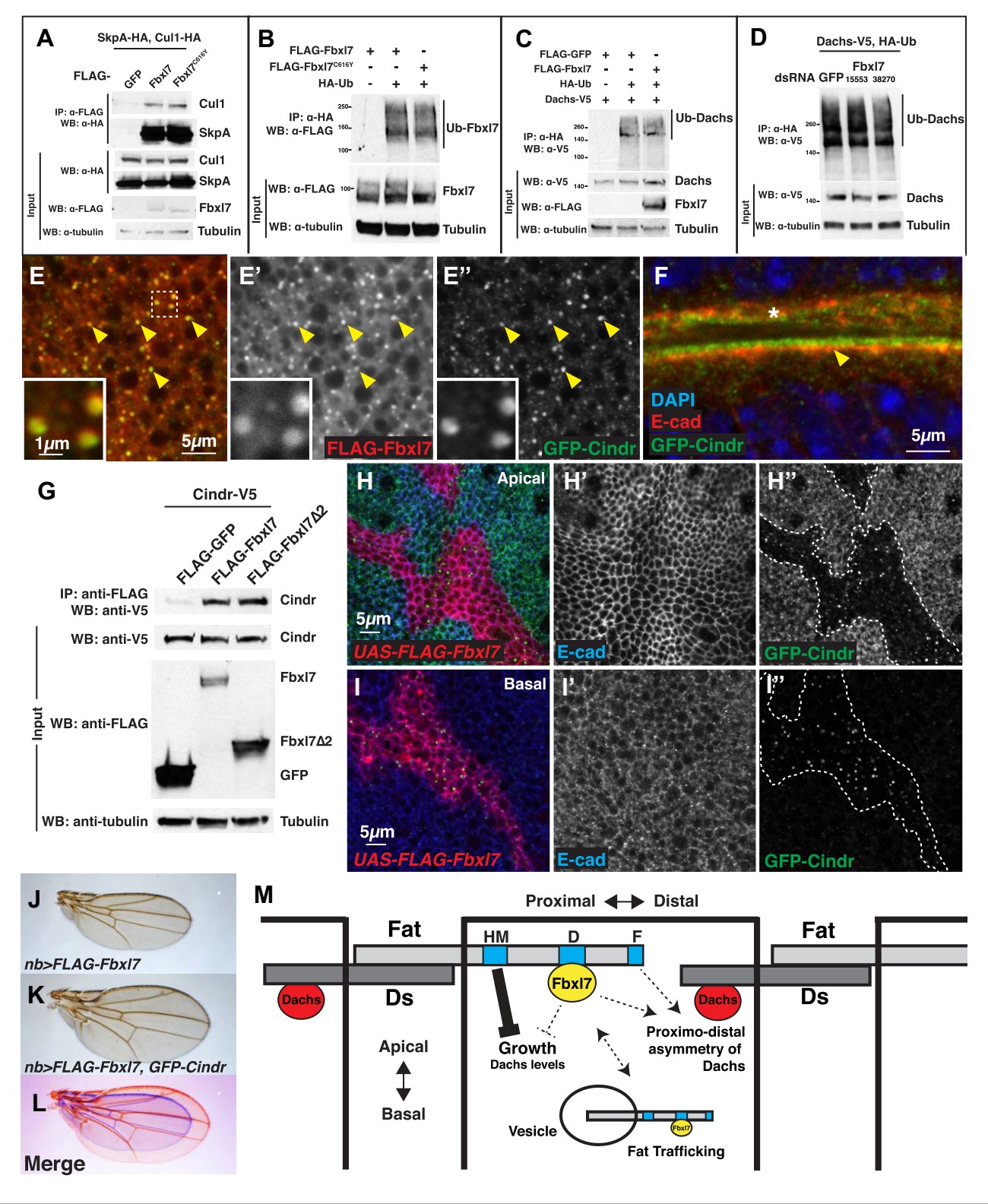

**Figure 7**. Fbxl7 does not affect Dachs ubiquitylation, and Fbxl7 affects the localization of Cindr. (**A**) Co-immunoprecipitation assay from S2 cells. SkpA-HA and Cul1-HA immunoprecipitates with FLAG-Fbxl7. (**B**) In-vivo Fbxl7 ubiquitylation assay in S2 cells. FLAG-Fbxl7 and FLAG-Fbxl7$^{C616Y}$ are ubiquitylated in vivo. (**C–D**) In-vivo Dachs ubiquitylation assay in S2 cells. Dachs-V5 is ubiquitylated under wild-type conditions, and does not change with (**C**) overexpression of FLAG-Fbxl7 or (**D**) knockdown of Fbxl7 with two different dsRNAs. (**E–E″**) Confocal slice showing localization of FLAG-Fbxl7

*Figure 7. Continued on next page*

*Figure 7. Continued*

(red) and GFP-Cindr (green) in puncta (arrowheads). (**F**) Confocal slice through a bend in the wing disc. GFP-Cindr (green) localizes to subapical membrane, apical to E-cad (red). Asterisk indicates an adjacent bend in the tissue. (**G**) Co-immunoprecipitation experiment in S2 cells. Cindr-V5 pulls down with full length FLAG-Fbxl7, and FLAG-Fbxl7Δ2, which contains only the LRR domains. (**H–I″**) Confocal slice in a disc with clones overexpressing FLAG-Fbxl7 (red, cells marked by myr-RFP) in a *GFP-Cindr* background. (**H–H″**) An apical plane shows loss of apical GFP-Cindr within the clone, and (**I–I″**) a basal plane shows accumulation of GFP-Cindr in puncta. (**J**) Compared to (**J**) *nb>FLAG-Fbxl7* alone, (**K**) overexpressing GFP-Cindr partially rescues the small wing phenotype. (**L**) Merge shows **J** blue and **K** in red. (**M**) Model of Fbxl7 as a component of Fat signaling. Not drawn to scale.

The following figure supplement is available for figure 7:

**Figure supplement 1**. Additional analysis of Dachs ubiquitylation, vesicle markers, and Cindr.

However, the level of ubiquitylation was unchanged when Fbxl7 was increased (*Figure 7C*, *Figure 7—figure supplement 1A*). Additionally, we reduced Fbxl7 in S2 cells by RNAi-mediated knockdown using two different dsRNAs and still observed no change in D ubiquitylation (*Figure 7D*). Importantly, we observe knockdown of Fbxl7 protein in S2 cells using these dsRNAs (*Figure 7—figure supplement 1B–C*). Furthermore, expressing a dominant negative version of Cul1 (Cul1$^{DN}$) does not impair D ubiquitylation (*Figure 7—figure supplement 1D*), implying that the SCF complex may not be involved. Thus we have no evidence that Fbxl7 influences D ubiquitylation.

## Fbxl7 co-localizes with Cindr and displaces it from the apical membrane

If Fbxl7 negatively regulates the accumulation of D at the apical membrane, it may do so by promoting the trafficking of D into intracellular vesicles. Indeed, we observed a population of intracellular puncta, likely vesicles, that contain both Fbxl7 and D (*Figure 6I–I″*). Moreover Fbxl7 overexpression can cause an overall shift in D from the apical membrane to the interior of the cell (*Figure 6I‴*). To further characterize the population of vesicles that contain Fbxl7, we examined the localization of FLAG-Fbxl7 with 59 different markers that each labeled a subpopulation of vesicles and with several proteins that have been identified as interactors of Ft in proteomic studies (*Kwon et al., 2013*) (*Supplementary file 1*). No co-localization was observed with most of these markers (*Figure 7—figure supplement 1E–G* as examples) and partial co-localization was seen with two markers of the retromer pathway, Snx3 and Vps35 (*Figure 7—figure supplement 1H–I″*). However, strong co-localization was seen with a protein-trap insertion of GFP in the *cindr* locus. Cindr is thought to be an adapter protein that links membrane proteins to the actin cytoskeleton ('Discussion'). In basal sections, there is almost complete overlap between GFP-Cindr and FLAG-Fbxl7 in puncta (*Figure 7E–E″*). GFP-Cindr is normally localized to the subapical membrane, apical to E-cadherin, but its localization there is less punctate and more diffuse than that of FLAG-Fbxl7 (*Figure 7F*). When tagged versions of both proteins were expressed in S2 cells, Cindr co-immunoprecipitated with full length Fbxl7 or with a version containing only the LRRs (Fbxl7Δ2) (*Figure 7G*).

To determine whether Fbxl7 could influence the cellular localization of GFP-Cindr, we overexpressed FLAG-Fbxl7 in clones in *GFP-Cindr* animals. In these clones, we observed a dramatic re-localization of GFP-Cindr. GFP-Cindr is almost entirely eliminated from the apical membrane (*Figure 7H–H″*) and increased numbers of basally located vesicles are observed (*Figure 7I–I″*). Thus, Fbxl7 is capable of displacing a protein associated with the apical membrane into intracellular vesicles. Importantly, this is unlikely to be a consequence of increased Ft in the apical membranes of Fbxl7-ovexpressing cells, since increasing Ft levels has no effect on the apical localization of GFP-Cindr (*Figure 7—figure supplement 1J–J″*). We next tested whether changes in Cindr levels are capable of modifying Fbxl7 phenotypes. Indeed we find that the reduction in wing size from overexpression of Fbxl7 was suppressed by co-expression of GFP-Cindr (*Figure 7J–L*, *Figure 7—figure supplement 1K*). Overexpression of GFP-Cindr alone causes slightly rounder wings with closer crossveins, though these wings were not significantly overgrown (*Figure 7—figure supplement 1L–O*).

Under conditions of Fbxl7 overexpression, we did not observe any increase in Cindr ubiquitylation indicating that Cindr is unlikely to be a direct target of Fbxl7 (not shown). Moreover, reducing Cindr levels by RNAi did not elicit phenotypic abnormalities in wings suggestive of defects in Ft or D (not shown). However, the ability of Fbxl7 to cause changes in the localization of Cindr and Ft indicates that it can regulate trafficking of proteins between the apical membrane and the interior of the cell in either direction, and the pathways that regulate the trafficking of these proteins

and D might share common components. Some of these shared components could potentially be direct targets of Fbxl7 ubiquitylation.

## Discussion

The protocadherin Ft lies at the apex of multiple pathways that together regulate growth, several aspects of PCP, and proximodistal patterning. The mechanism by which Ft functions as a signaling molecule remains poorly understood. We have now identified the F-box protein Fbxl7 as an immediate effector of Ft, that functions to restrict the levels of the atypical myosin D at the apical membrane as well as its distribution around the perimeter of the cell. In addition, Fbxl7 can regulate levels of Ft at the apical membrane.

### Multiple effector pathways downstream of fat

Recent studies have revealed that Ft's effects on distinct pathways may be genetically separated, and that multiple effector domains can contribute to the same function. Indeed, the growth-suppressing function of Ft may occur via at least two regions of the Ft ICD. One or more regions between amino acids 4834 and 4899 in full-length Ft appear responsible for Ft's ability to regulate Hippo signaling (labeled HM in *Figure 7M*) (*Matakatsu and Blair, 2012*; *Bossuyt et al., 2013*; *Zhao et al., 2013*). Several mutations within this region compromise this function of Ft and cause massive tissue overgrowth (*Bossuyt et al., 2013*). Intriguingly, an allele of *ft*, *ft61*, which harbors such a mutation, showed neither an effect on the recruitment of Fbxl7 to the apical membrane nor on the binding of Ft to Fbxl7. Thus, signaling via this region of the ICD appears to be independent of Fbxl7. A second, more C-terminal region of the Ft ICD (Region D in *Figure 7M*) that extends between amino acids 4975 and 4993 of full-length Ft, is removed by the *ftΔD* deletion and also has a growth-suppressive function albeit weaker than that of HM (*Pan et al., 2013*). This second growth-suppressive pathway requires the function of Fbxl7, as the protein generated by the *ftΔD* allele cannot bind to Fbxl7 nor can it localize Fbxl7 to the apical membrane. Additionally, the phenotypic abnormalities of null alleles of *ft* rescued by *ftΔD* are very similar, if not identical to those of *Fbxl7* mutants. Furthermore, like *ftΔD*, *Fbxl7* mutations do not display overt abnormalities of hair orientation in the wing (*Figure 1—figure supplement 2E–J*), or abdomen (not shown).

We have shown that hyperactivation of the "weaker" Fbxl7-dependent pathway can overcome the absence of the 'stronger' Fbxl7-independent pathway; overexpression of Fbxl7 can suppress the overgrowth of *ft61*. Thus, while these two pathways can be dissociated at the level of the Ft ICD, they nevertheless seem to converge further downstream. This point of convergence likely involves D since the overgrowth of *ft* mutant tissue can be suppressed completely by eliminating D function (*Cho et al., 2006*). Indeed, it has previously been suggested that Ft regulates growth by restricting the levels of apical D, and regulates PCP by influencing the planar asymmetry of apical D (*Rogulja et al., 2008*; *Pan et al., 2013*).

Another key finding in our experiments is that *Fbxl7* mutations perturb the distribution of D around the perimeter of the apical region of the cell. D is normally biased towards the distal edge of the cell; in *Fbxl7* mutants, D is more evenly distributed around the cell perimeter. The asymmetric localization of D depends on at least two different regions of Ft (*Pan et al., 2013*). One is the region that binds to Fbxl7 (Region D) and the other is composed of the last three amino acids at the C-terminus of the protein (Region F in *Figure 7M*), which is not necessary for Fbxl7 localization to the apical membrane. Thus, for the regulation of D asymmetry as well, there appears to be an Fbxl7-independent pathway. The existence of multiple downstream effector pathways that converge on common biological outcomes suggests that these pathways might function redundantly to some extent and thus provide robustness. This might also explain why the phenotypes elicited by overexpression of Fbxl7 are, in general, more severe than those observed in loss-of-function mutations.

### Fbxl7 as a regulator of protein localization

Previous observations of the localization of Ft, Ds, and D to vesicles are suggestive of trafficking events being involved in Ft signaling (*Ma et al., 2003*; *Matakatsu and Blair, 2004*; *Mao et al., 2006*). We have demonstrated that, in addition to the apical membrane, Fbxl7 localizes to vesicles. Moreover, FLAG-Fbxl7 vesicles can contain Ft, Ds and D, and these may be related to the apical puncta observed on cell edges. This localization is likely specific, since we do not see Fbxl7 co-localization with other

cell surface proteins such as Crumbs, Notch, and E-cadherin (not shown). Currently very little is known about the role of each of these proteins in vesicles. However, there is an increasing appreciation that most transmembrane proteins, and even proteins that are associated with the inner leaflet of the cell membrane are maintained at the plasma membrane by a dynamic process involving endocytosis and vesicle recycling (e.g., *Schmick et al., 2014*).

We provide evidence that Fbxl7 regulates Ft apical localization, but how this regulation relates to the Fbxl7 phenotypes is not clear. Since Fbxl7 overexpression increases Fat signaling, and rescues the overgrowth-inducing Ft[61] allele, perhaps this is due to the increased levels of Ft protein at the apical membrane. However, Ft levels are slightly elevated in *Fbxl7* mutants, which display mild overgrowth. Therefore the mutant phenotype cannot be explained by the effect on Ft. Another known regulator of apical Ft levels is *lowfat* (*lft*) (*Mao et al., 2009*). Fbxl7 and Lft appear to regulate Ft in different ways. Lft overexpression, like Fbxl7, increases Ft levels. However, while Ft levels are decreased in *lft* mutant cells, Ft levels are increased in *Fbxl7* mutant cells, though less so compared to *Fbxl7* overexpression. Interestingly, for many proteins that regulate cellular trafficking, similar phenotypic abnormalities are observed with gain-of-function and loss-of-function mutations, since the normal execution of the process requires the protein to shuttle efficiently between two states (*Park et al., 1993*). Thus dynamic aspects of the localization of Ft, Ds and D clearly merit more attention.

The interactions we have observed between Fbxl7 and the adapter protein Cindr may provide clues for how Fbxl7 regulates D localization. Fbxl7-associated vesicles show almost complete overlap with GFP-Cindr and Fbxl7 can re-localize Cindr from the apical membrane to the interior of the cell. This finding, together with the observed increase in basal levels of D upon Fbxl7 overexpression (*Figure 6I–I‴*), suggests that Fbxl7 may function to regulate D trafficking in a similar manner. Cindr and its mammalian orthologues Cin85 and CD2AP are thought to regulate interactions between membrane proteins and actin cytoskeleton (*Haglund et al., 2002*; *Petrelli et al., 2002*; *Soubeyran et al., 2002*; *Johnson et al., 2011*, *2012*). D is an atypical myosin with a predicted actin binding domain in its conserved head domain. Therefore, the vesicles which Fbxl7 associates with D and Cindr may be linked to the actin cytoskeleton. In addition, our finding of partial colocalization of Fbxl7 with retromer components further supports the possibility that Fbxl7 may have a role in protein trafficking.

## Fbxl7 as a ubiquitin-ligase component

Many F-box proteins associate with Skp1 and Cul1 to form an SCF E3 ubiquitin ligase complex (reviewed in *Skaar et al., 2013*). Recruitment of specific substrates results in their poly-ubiquitylation and degradation, or mono-ubiquitylation, which can have non-degradative signaling roles. In addition, some F-box proteins have SCF-independent roles (*Nelson et al., 2013*). Fbxl proteins are thought to recruit substrates to the SCF complex through the interaction with their LRR domains, and substrates have been identified for several Fbxls such as Skp2 (Fbxl1), which degrades p27 (*Carrano et al., 1999*; *Sutterluty et al., 1999*). However many, like Fbxl7, are still uncharacterized as 'orphan' F-box proteins with no known substrates.

Since we find that Fbxl7 associates with Skp1 and Cul1, its potential substrates may be involved in Ft signaling. Fbxl7 has one described substrate in mice, Aurora A (*Coon et al., 2012*). However we do not believe Aurora A is a relevant substrate in *Drosophila*, as we do not observe Ft signaling defects when Aurora A is knocked down or overexpressed (not shown). The identification of F-box protein substrates has mainly been accomplished by unbiased approaches (*Skaar et al., 2013*). Similarly, a combination of unbiased approaches, involving proteomics, genetic interaction screens, and identifying proteins that co-localize with Fbxl7 in vesicles could be used to identify Fbxl7 substrates.

# Materials and methods

## *Drosophila* genetics

*Fbxl7[C616Y]* and *Fbxl7[Q201X]* alleles were isolated in two EMS screens, *Fbxl7[W389X]* was found fortuitously in a separate fly stock, and *Fbxl7[MI04292]* (BL37813) is a *MI{MIC}* insertion in the first intron of *Fbxl7*. All *Fbxl7* alleles are on chromosomes bearing a *FRT82B* insertion. *Fbxl7* overexpression stocks used were *UAS-FLAG-Fbxl7* (this study, attP40 and attP2), *UAS-FLAG-Fbxl7[C616Y]* (this study, attP40),

and P{XP}CG4221*d08178* (BL19289). *Fbxl7* RNAi stocks used were *UAS-Fbxl7RNAi* (JF01515 [BL31065], VDRC108628). All RNAi experiments performed in flies used UAS-Dcr2, which increases knockdown. The *fat61* allele was isolated in an EMS screen for supercompetitor mutations (T4854I amino acid change).

Additional stocks used were: FRT82B *dcole88* (**Jursnich et al., 1990**), P[acman]-Fat+; P[acman]-FatΔD, P[acman]-FatΔF (**Pan et al., 2013**), Diap1 3.5-GFP (**Zhang et al., 2008**), FRT42D *fjN7*, FRT40A *ftGrv*, FRT40A *ds38k*, UAS-Fat (**Simon, 2004**), *ykiB5* (**Huang et al., 2005**), Tub-EGFP.ban ('bantam sensor', **Brennecke et al., 2003**), FRT40A, FRT82B (**Xu and Rubin, 1993**), UAS-GFP-cindr-PC (**Johnson et al., 2008**), Dachs-GFP (**Bosveld et al., 2012**).

Remaining stocks used were from, or derived from, the Bloomington Stock Center (Bloomington, IN): UAS-dcr2; nub-Gal4 (BL25754), eyFLP; FRT82B ub-GFP (BL5580, BL5188), FRT82B ub-RFPnls (BL30555) hsFLP;; Act>CD2>Gal4 UAS-GFP (BL26902, BL4780), FRT82B MARCM (BL30036), FRT40A MARCM (BL5192), FRT40A *exe1* (BL44249), FRT82B *wtsX1* (BL44251), FRT82B *dco3* (BL44250), FRT40A *dGC13* (BL28289), UAS-d:v5 (BL28291), UAS-zyx-ChRFP (BL28875), UAS-fj:V5 (BL44252), Df(3R)BSC515 (BL25019), Df(3R)BSC728 (BL26580), GFP-Cindr*CA06686* (BL50802), act-Gal4 (BL3954), tub-Gal4 (BL5138), FRT40A *ft8* (BL44257), hs-Gal4 (BL1799), UAS-EGFP (BL6658), UAS-myr-mRFP (BL7119), UAS-GFP-KDEL (BL9898), UAS-Galt-GFP (BL30902), UAS-GFP-myc-2xFYVE (BL42712), eyFLP; 40A CL white+/ CyO (BL5622), en-Gal4 UAS-RFP (BL30557).

Stocks used for vesicle co-localization are listed in *Supplementary file 1* and are listed with BL numbers if available.

## Tissue immunohistochemistry

*hsFLP*-induced clones were generated by incubating larvae at 37°C at 48 hr after egg deposition (AED). A 30-min incubation was used for experiments using *Act>CD2>Gal4* and 2-hr incubation for experiments using *MARCM*. Immunostainings were performed by dissecting imaginal discs from wandering third instar larvae, fixing discs in 4% paraformaldehyde + PBS, followed by blocking in PBS + 0.1% Triton-X + 5% normal goat serum (NGS), incubation with primary antibodies overnight at 4°C, and incubation with secondary antibodies overnight at 4°C. Immunostainings with anti-Fbxl7 antibodies required a separate optimized protocol: Larvae were dissected in 0.1 M NaPO$_4$, fixed in PLP-fixative (2% paraformaldehyde, 0.01 M NaIO$_4$, 0.075 M lysine, 0.037 M NaPO$_4$), washed with 0.1 M NaPO$_4$ containing 0.1% saponin, blocked with 0.1 M NaPO$_4$ containing 0.1% saponin and 5% NGS, primary and secondary antibodies were diluted in 0.1 M NaPO$_4$ containing 0.1% saponin and 5% NGS. Samples were imaged on a Zeiss 700 confocal microscope (Germany).

The anti-Fbxl7 antibody was generated by immunizing guinea pigs (Pocono Farms, Canadensis, PA) with purified Fbxl7 (amino acids 22–324) produced at the UC-Berkeley MacroLab (His-Fbxl7 purified on a Nickel column), and used at 1:1000 for tissue staining.

Other antibodies used: rat anti-Dachs (1:500, **Brittle et al., 2012**), rat anti-Fat (1:1600, **Feng and Irvine, 2009**), rat anti-Dachsous (1:5000, **Yang et al., 2002**) rat anti-Ecad (1:100, DCAD2, DHSB, Iowa City, IA), mouse anti-FLAG (1:1000, F3165; Sigma, St. Louis, MO), rabbit anti-FLAG (1:1000, F7425; Sigma) mouse anti-V5 (1:500, R960-25; Invitrogen, Carlsbad, CA), mouse anti-Arm (1:100, N2 7A1; DHSB), rabbit anti-LacZ (1:500, #559762; MP Biomedicals, Santa Ana, CA), anti-Cleaved Caspase-3 (1:200, 9661; Cell Signaling, Beverly, MA). Actin was visualized with Phalloidin-TRITC (1:500, Sigma) or Alexa Fluor 633 Phalloidin (1:500, Invitrogen). Nuclei were visualized with DAPI (1:1000).

## Plasmids and molecular biology

Plasmids were constructed using conventional ligation-based molecular cloning techniques. Oligonucleotide sequences are listed in a separate table in *Supplemental file 2*. Fbxl7 was amplified from clone LD38495 (DGRC, Bloomington, IN) by designing oligonucleotides to amplify the single predicted coding sequence CG4221-RA and add Not1 and Xba1 restriction sites. The Not1-Fbxl7-Xba1 PCR fragment was digested and ligated into pUAS-FLAG attB (adds an N-terminal FLAG tag) to generate pUAS-FLAG-Fbxl7 attB. The C616Y amino acid change was introduced by site directed mutagenesis, generating pUAS-FLAG-Fbxl7*C616Y* attB. Transgenic flies were made from pUAS-FLAG-Fbxl7 attB and pUAS-FLAG-Fbxl7*C616Y* attB using PhiC31 integration (BestGene, Chino Hills, CA), inserting into attP40 and attP2 landing sites.

Fbxl7 truncation plasmids were generated by amplifying Fbxl7Δ1 (389-772aa), Fbxl7Δ2 (445-772aa), and Fbxl7Δ3 (1-388aa) and ligating into pUAS-FLAG attB using Not1/Xba1. pUAS-FLAG-EGFP

attB was generated by amplifying EGFP from pEGFPattB (K Basler) and cloning into pUAS-FLAG attB using In-Fusion (Clonetech, Mountain View, CA).

SkpA and Cul1 coding sequence were amplified from genomic DNA and clone LD20253 (DGRC), respectively. Not1/Xba1 sites were added to oligos that amplified SkpA, and Kpn1/Not1 was added for Cul1. PCR fragments were digested and ligated into pMT-HA (adds a C-terminal HA tag), generating pMT-SkpA-HA and pMT-Cul1-HA. dCul1[DN] is a C-terminal truncation (1-451aa) which corresponds to 1-452aa of dominant negative human hCul1DN (*Wu et al., 2000*) and was cloned into pMT-HA as for full length dCul1.

pMT-FatICD-V5 was generated by amplifying FatICD coding sequence from pUAS-FatICD-V5 (K. Irvine), adding Not1/Xba1 sites with oligos. PCR fragments were digested and ligated into pMT-V5/6xHis (Invitrogen). pMT-FatICDΔD-V5, pMT-FatICDΔF-V5, and pMT-FatICDmutV-V5 were generated by using the same oligos to amplify from pUAS-FatICDΔD-V5, pUAS-FatICDΔF-V5, and pUAS-FatICDmutV-V5 (Irvine), respectively. pMT-FatICD61-V5 was generated by site directed mutagenesis of pMT-FatICD-V5 to make the change T4854I.

pUAS-HA-Ub attB was generated by amplifying Ubi-p5E coding sequence from genomic DNA, adding an N-terminal HA tag with primers, and inserting into pUAS attB (K Basler).

pMT-cindr-V5 was generated by amplifying the longest predicted isoform cindr-RC from S2R+ cell cDNA, adding Not1/Xba1 sites, and ligating into pMT-V5/6xHis (Invitrogen).

Other plasmids used are pMT-Dco-V5 (*Ko et al., 2002*), pUAS-Dachs-V5 (*Mao et al., 2006*).

The sequence of oligos used are in *Supplementary file 2*.

## Western analysis of wing discs and S2R+ cells, co-immunoprecipitation, and in vivo ubiquitylation assays

S2R+ cells were cultured and transfected using conventional techniques. S2R+ cells were cultured in Schneiders medium containing 10% FBS at 27°C, transfected with Effectene (Qiagen, Germany) in six-well dishes, and harvested 72 hr later. 500 µM CuSO4 was added 24 hr before harvesting to induce expression from plasmids with metalothionein promoters. For Co-IP and in vivo ubiquitylation assays, 50 µM MG132 (C2211; Sigma) was added to transfected cells four hours before harvesting to inhibit proteasome activity. For experiments using dsRNA, S2R+ cells were transfected with dsRNA ± plasmids and were harvested as needed for protocols described above.

Unless otherwise stated, wing discs or S2R+ cells were boiled in 1x or 2x SDS Sample buffer, run on 7.5% Mini-Protean TGX gels (Bio-Rad, Hercules, CA), and transferred to nitrocellulose membrane. Protein bands were detected with primary antibodies and secondary antibodies conjugated to HRP, and imaged using ECL detection reagent (RPN2232; Amersham, UK).

For co-IP assays, 50 µM MG132 (C2211; Sigma) was added to transfected cells 4 hr before harvesting to inhibit proteasome activity. Cells were washed once with ice cold PBS, and lysed in lysis buffer (20 mM HEPES 7.5 pH, 5 mM KCl, 1 mM MgCl2, 0.1% NP-40, 'Complete' EDTA free protease inhibitor tablet [Roche, Switzerland]). Insoluble material and nuclei were removed by centrifugation at 13,000×$g$ for 30 min at 4°C, and soluble cell lysate was incubated with anti-FLAG M2 affinity gel (A2220; Sigma) overnight at 4°C. Beads were washed twice in lysis buffer and denatured by boiling in SDS sample buffer for 10 min. For SkpA, Cul1, and Cindr co-IP assays, to avoid detection of non-specific binding of transfected proteins to beads, FLAG-protein complexes were eluted off beads by incubating with 400 ng/µl 3x FLAG peptide (F4799; Sigma) for 30 min at 4°C.

For in vivo ubiquitylation assays, 50 µM MG132 was added to transfected cells 4 hr before harvesting. Cells were washed once with ice cold PBS, and proteins denatured by boiling in 100 µl 1% SDS in PBS for 10 min 400 µl of 0.5% BSA, 1%Triton-X, in PBS was added, and samples were sonicated, then centrifuged at 13,000×$g$ for 10 min. Supernatant was diluted to 1 ml with 5% BSA, 1%Triton-X and incubated with anti-HA agarose beads (A2095; Sigma) overnight at 4°C. Beads were washed twice with 1%Triton-X in PBS and boiled in SDS sample buffer for 10 min. For in vivo ubiquitylation of Dachs-V5 from larval tissue, 12 hr before dissection larvae were heat-shocked at 37°C for 1 hr to induce UAS transgenes by hs-Gal4. 30 brain-eye-antennal complexes per genotype were dissected in Schneiders medium and incubated with 50 µM MG132 for 4 hr. Complexes were boiled, diluted, sonicated, and centrifuged as above. Supernatant was diluted to 1 ml with 5% BSA, 1%Triton-X and incubated with Protein G Sepharose (P3296; Sigma) for 1 hr at 4°C, replaced with Protein G Sepharose plus 1 µl mouse anti-V5 antibody (R960-25; Invitrogen) and incubated overnight at 4°C. Beads were washed twice with 1%Triton-X in PBS and boiled in SDS sample buffer for 10 min.

For experiments using dsRNA, S2R+ cells were transfected with dsRNA ± plasmids and were harvested as needed for protocols described above. dsRNA was generated by PCR amplifying DRSC15513 and DRSC38270 from genomic DNA, and GFP coding sequence from pattB-EGFP (K Basler), adding T7 sequence to forward and reverse primers, and in vitro transcribing dsRNA (AM1333; Megascript T7 Transcription Kit, Invitrogen).

For anti-Fat western blots from wing discs, 20 wing discs were dissected from third instar larvae in PBS and immediately boiled in 2x SDS Sample buffer. The amount loaded on gels was adjusted to load equivalent amounts of protein.

For anti-Dachs westerns from wing discs, 20 wing discs were dissected from third instar larvae in PBS and lysed in 1x RIPA buffer. Total protein was quantified (Micro BCA kit, 23235; Fisher, Hampton, NH) and adjusted equally among samples. Secondary antibodies conjugated to LiCor fluorescent dyes were used to detect protein bands using a LiCor Odessey imager (Lincoln, NE).

Western blots were probed with the following antibodies:

Guinea pig anti-Fbxl7 (1:1000), rat anti-Fat (1:25,000, K Irvine), rat anti-Dachs (1:5,000, D Strutt), mouse anti-Tubulin (1:100, E7; DHSB), mouse anti-FLAG (1:10,000, F3165; Sigma), mouse anti-V5 (1:5,000; R960-25, Invitrogen), rabbit anti-V5 (1:5000, V8137; Sigma) rabbit anti-HA (1:1,000, 3724; Cell Signaling), rabbit anti-Ub (1:1,000, Z0458; DakoCytomation, Carpinteria, CA), mouse anti-Lamin (1:100, ADL67.10; DHSB), goat anti-rat-HRP (112-035-003; Jackson, West Grove, PA), goat anti-rabbit-HRP (111-035-003; Jackson), goat anti-mouse-HRP (172-1011; BioRad), goat anti-guinea pig-HRP (106-035-003; Jackson), goat anti-rat-IR680 (926-68,076; Licor), goat anti-mouse-IR800 (827-08,364; Licor).

## Quantification of dimensions of adult structures

Wings or legs were mounted onto slides using Canadian Balsam medium (Gary's Magic Mount) and imaged on a Leica transmitted light microscope (TL RCI, Germany). Wing area and cross vein distance was quantified in ImageJ. For cross veins, we measured the distance of a straight line drawn from intersection of the anterior cross vein and L4 to the intersection of the posterior cross vein and L4. Statistical significance between groups was determined by one-way ANOVA using (Tukey's or Dunnett's test).

## Quantification of Dachs asymmetry in wing discs

Quantifications were performed as in *Brittle et al. (2012)* using ImageJ. Wing discs were immunostained for Dachs and F-actin and imaged under identical settings at 20x to determine P-D orientation, and at 63x to image the dorsal portion of the wing pouch where Dachs asymmetry is highest. Images were rotated so that the P-D axis of the wing pouch oriented vertically (90° and 270°). A cropped 24.8 × 24.8um (500 × 500px) square was used to quantify the mean fluorescence intensity of Dachs or actin along each cell edge while recording the angle of the cell edge relative to the P-D orientation. Cell edge data were measured using a 1 pixel width line. Mean fluorescence of cell edges oriented in the P-D orientation (45°–135°) or the A-P orientation (0°–45°; 135°–180°) was isolated into two different lists, which were each averaged. The ratio of mean fluorescence of the A-P orientation to P-D orientation gives the P-D/A-P localization. For example, asymmetric localization to the P-D sides of cells will give higher mean intensities on cell edges in the A-P orientation. Quantifications were performed on eight cropped boxes from different discs for each group. Statistical significance between groups was determined by one-way ANOVA using (Tukey's test).

## Acknowledgements

We thank K Basler, D Bilder, S Blair, J Brill, L Cooley, R Fehon, D Strutt, K Irvine, J Jiang, R Johnson, H Kramer, H McNeill, J Price, M Simon, and M Welch for stocks and reagents. We thank D Bilder, G Garriga and M Rape for advice. We thank A Figueroa-Clarevega and S Rosemond for help with screening and C Van for help with plasmid construction. We thank the Bloomington Stock Center, *Drosophila* Genomics Resource Center, and Developmental Studies Hybridoma Bank, and UC-Berkeley MacroLab for stocks, reagents and services. This work was supported by grant R01GM61672 from the National Institutes of Health (NIGMS) and an American Cancer Society Research Professor Award 120366-RP-11-078-01-DDC to IKH. JAB was funded by a National Science Foundation graduate research fellowship and then a fellowship from the Cancer Research Coordinating Committee of the University of California.

# Additional information

## Funding

| Funder | Grant reference number | Author |
|---|---|---|
| National Institute of General Medical Sciences | R01 GM61672 | Iswar K Hariharan |
| American Cancer Society | 120366-RP-11-078-01-DDC | Iswar K Hariharan |
| National Science Foundation | | Justin A Bosch |

The funders had no role in study design, data collection and interpretation, or the decision to submit the work for publication.

## Author contributions

JAB, TMS, Conception and design, Acquisition of data, Analysis and interpretation of data, Drafting or revising the article; YH, BJP, Conducted screens that generated Fbxl7 alleles; KDG, Helped with mapping and sequencing mutant alleles and phenotypic analysis; IKH, Conception and design, Analysis and interpretation of data, Drafting or revising the article

## Ethics

Animal experimentation: Custom antibodies used in this study were generated in guinea pigs by the vendor Pocono Rabbit Farm and Laboratory. Pocono Rabbit Farm and Laboratory, Inc. has an Animal Welfare Assurance on file with The Office of Laboratory Animal Welfare (OLAW). The Animal Welfare Assurance number is A3886-01. This study was performed in strict accordance with the recommendations in the Guide for the Care and Use of Laboratory Animals of the National Institutes of Health.

# Additional files

## Supplementary files

• Supplementary file 1. Localization of FLAG-Fbxl7 and vesicle markers. List of tagged proteins that localize to vesicles, are related to vesicle trafficking, and proteins interacting with Fat. Colocalization was assessed with FLAG-Fbxl7 in the wing imaginal disc.

• Supplementary file 2. Sequence of oligonucleotides used.

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
