## [Decision Letter]

Thank you for sending your work entitled “The Drosophila F-box protein Fbxl7 associates with the protocadherin Fat and regulates signaling via the Hippo pathway” for consideration at *eLife*. Your article has been favorably evaluated by K VijayRaghavan (Senior editor) and 3 reviewers, one of whom is a member of our Board of Reviewing Editors.

This manuscript describe Fbxl7 as a new component of the Hippo signalling pathway downstream of Fat, thus providing important insights into how this transmembrane protein signals to control growth. They investigate relationships between several proteins including Fat, Dachsous, Dachs, Dco and vesicle trafficking proteins such as Cindr. The experiments are very rigorous and thorough, employing multiple mutant alleles, RNAi lines, transgenes, antibodies and biochemical reagents. It seems quite clear that Fbxl7 functions downstream of Fat to regulate growth and controls the apical localization and abundance of Dachs, although its mechanistic links to Dco and Cindr are not entirely clear. The manuscript represents a huge amount of work and while there are some loose ends mechanistically, substantial efforts have been made to make conclusions without over-interpreting data or forcing results.

The reviewers all agreed that the work described in this paper was novel, important and appropriate for publication in *eLife*. The reviewers had a few experimental points that they felt would strengthen the paper, and several points that they felt could be improved/clarified in the Discussion before publication.

Experimental/image issues:

1) The Western blots in Figure 6 are not great quality. The reviewers appreciate that there might be some technical issues with the Dachs antibody but if it can be improved it will make this result clearer and add value to the manuscript. At present, these Western blots look like they have been over-modified with software such as Adobe Photoshop.

2) Figure 5: The staining of loss of Fbxl7 in *ft*^*GrV*^*/ft*^*8*^*; ftΔD* is not very clear compared to controls in D. Can this be improved?

3) Can fat loss upregulate and mislocalize Dachs in the absence of Fbxl7? The authors suggest that Fbxl7 has a role mediating some of Fat's effects on Dachs levels and asymmetry, likely by binding Fat's D domain. They also argue that the F domain's effects on Dachs asymmetry are not mediated by Fbxl7, but this is only based on the finding that removing the F domain does not detectably change Fbxl7 binding. However, it is still possible that the F domain's effects are mediated by Fbxl7, especially since Fbxl7 has residual binding to Fat lacking the D domain. It would be helpful if they tested how important Fbxl7 is for Fat's effects on Dachs. Can a form of Fat deficient in Fbxl7 binding, such as ΔD, still improve Dachs in a fat mutant?

4) The authors do not present any evidence that the D domain is required for Fbxl7's effects on Fat levels. Does Fbxl7 (gain or loss) affect the apical levels of FatΔD?

DISCUSSION POINTS

1) Fat localization: The authors show a very strong (although complex) role of Fbxl7 controlling Fat localization. However, this role hardly gets mentioned in the Discussion, and so it is not at all clear how and if it connects to the Fbxl7 phenotype. In the Results, in contrast, the authors suggest that some of the effects of Fbxl7 are “amplified” by changing Fat levels, and that this effect is likely mediated through the D domain. More Discussion should be devoted to the alterations in Fat levels. A brief comparison to Lowfat is also probably in order, as Lowfat also regulates Fat levels.

2) Fbxl2 overexpression: This causes a phenotype very similar to that of Dachs mutants, but only causes very subtle effects on Dachs localization. If much or all of Fbxl2's effects are mediated via Dachs, as the authors suggest, this requires a little explanation.

3) I don't think we can conclude that Fbxl7 function is not related to the growth-suppressive function that is perturbed in *ft*^*61*^ alleles, just that its localization is not affected by the ft61 mutation. Indeed Fbxl7 function could be disrupted in ft61 alleles; perhaps a more likely scenario is that Fat61 protein binds to Fbxl7 but cannot modulate its activity because it fails to recruit an accessory protein that either modulates Fbxl7 activity or that is regulated by Fbxl7.

4) The text relating to the blots in Figure 6—figure supplement 1 states that Dachs levels do not go up when Fbxl7 is increased but they actually do look higher. This also seems to hold true in Figure 7. However, in Figure 7, Fbxl7 RNAi does not seem to affect Dachs levels.

5) The final model figure is not useful, and is at times misleading. Particularly, the labeling of the regions 1,2 and 3 is not helpful, as it adds yet another unmatching set of designations to the Ft cytoplasmic domain. It would be more useful if the authors adapted Figure 5 to their final model. Also, the arrows to Dachs might be better if they were dotted, or via question marks, or possibly across cell membranes to Ds, as all the data in their paper indicates that the link to Dachs is indirect.

6) The manuscript gets a bit bogged down with the negative data about the possible type of vesicles, possible functions of Dachs and the inclusion of the Cindr data. The manuscript would be improved if this section of the paper was reduced, as it is largely negative data. Similarly, it would be more useful if the Discussion addressed what the authors have shown and their model, and reduce the discussion of vesicles and Cindr, as the link to the title and the Abstract is not carried through.

---

## [Author Response]

Experimental/image issues:

*1) The Western blots in*
Figure 6
*are not great quality. The reviewers appreciate that there might be some technical issues with the Dachs antibody but if it can be improved it will make this result clearer and add value to the manuscript. At present, these Western blots look like they have been over-modified with software such as Adobe Photoshop*.

The signals obtained in Western blots using the anti-Dachs antibody are always weak. We have done the experiment many times to satisfy ourselves that the results are consistent – we never see a change in Dachs protein levels in Fbxl7 mutant discs, though the blots are never pretty because of the poor level of signal above background. We have replaced the blot used in the original Figure 6 with another one. That original blot is now in the supplemental figure (with less modification of brightness and contrast), we have retained it in the supplemental figure because it shows discs of additional genotypes as well.

*2)*
Figure 5*: The staining of loss of Fbxl7 in* ft^GrV^/ft^8^; ftΔD *is not very clear compared to controls in D. Can this be improved?*

We had previously shown only confocal slices through a bend in the wing disc, which is used to show apically localized proteins. We now show the x-y images, and z-sections, which show very clearly that the levels at the apical membrane are reduced considerably.

3) Can fat loss upregulate and mislocalize Dachs in the absence of Fbxl7? The authors suggest that Fbxl7 has a role mediating some of Fat's effects on Dachs levels and asymmetry, likely by binding Fat's D domain. They also argue that the F domain's effects on Dachs asymmetry are not mediated by Fbxl7, but this is only based on the finding that removing the F domain does not detectably change Fbxl7 binding. However, it is still possible that the F domain's effects are mediated by Fbxl7, especially since Fbxl7 has residual binding to Fat lacking the D domain. It would be helpful if they tested how important Fbxl7 is for Fat's effects on Dachs. Can a form of Fat deficient in Fbxl7 binding, such as ΔD, still improve Dachs in a fat mutant?

We are confused by the first sentence of this comment since loss of either Fbxl7 or loss of Fat already results in Dachs mislocalization. If the reviewers were wondering whether overexpression of Fbxl7 can rescue some of the Dachs defects in *ft* mutants, we have now added data to show that the increased Dachs levels in *ft*^*61*^ clones are suppressed by increasing Fbxl7 levels (Figure 6—figure supplement 1). Please note that the Ft^61^ protein can still bind to and apically localize Fbxl7.

The reviewers also point out that the D-deletion does not completely abolish Fbxl7 binding to Ft and suggest that there might be residual binding via the F-domain. We have included additional data (Figure 5—figure supplement 2) that show that overexpression of Fbxl7 increases the apical levels of wild-type Ft and FtΔF but not FtΔD suggesting that Fbxl7 needs to bind to Ft via the D domain but not the F domain (at least by this assay). We think the presence of a low level of binding in the ΔD mutant may simply be that the D deletion only partially removes the site that binds Ft and may therefore reduce the binding affinity substantially. In the absence of structural information, this issue cannot be fully resolved.

It would be helpful if they tested how important Fbxl7 is for Fat's effects on Dachs. Can a form of Fat deficient in Fbxl7 binding, such as deltaD, still improve Dachs in a fat mutant?

Work published by the Irvine lab (46) shows that the FtΔD mutant provides partial rescue of the Dachs localization defect. We have cited this work.

4) The authors do not present any evidence that the D domain is required for Fbxl7's effects on Fat levels. Does Fbxl7 (gain or loss) affect the apical levels of FatΔD?

Thank you for suggesting this experiment. We have included additional data (Figure 5—figure supplement 2) that show very clearly that the D domain is necessary for observing an increase in apical Ft levels upon Fbxl7 overexpression and that the F domain is not.

DISCUSSION POINTS

*1) Fat localization: The authors show a very strong (although complex) role of Fbxl7 controlling Fat localization. However, this role hardly gets mentioned in the Discussion, and so it is not at all clear how and if it connects to the Fbxl7 phenotype. In the Results, in contrast, the authors suggest that some of the effects of Fbxl7 are “amplified” by changing Fat levels, and that this effect is likely mediated through the D domain. More Discussion should be devoted to the alterations in Fat levels. A brief comparison to Lowfat is also probably in order, as Lowfat also regulates Fat levels*.

We have added discussion of this issue. We emphasize that some of the effects of Fbxl7 overexpression could be mediated by an increase in the levels of apical Ft. However, the phenotypic abnormalities observed in Fbxl7 loss-of-function mutants are not likely to result from the small increase in apical Ft levels. We have included a comparison to Lowfat as suggested by the reviewers.

*2) Fbxl2 overexpression: This causes a phenotype very similar to that of Dachs mutants, but only causes very subtle effects on Dachs localization. If much or all of Fbxl2's effects are mediated via Dachs, as the authors suggest, this requires a little explanation*.

We have included additional data using a Dachs-GFP, which in combination with the images using the anti-Dachs antibody, shows very clearly that Dachs levels at the apical membrane are reduced when Fbxl7 levels are increased.

*3) I don't think we can conclude that Fbxl7 function is not related to the growth-suppressive function that is perturbed in* ft^61^
*alleles, just that its localization is not affected by the ft61 mutation. Indeed Fbxl7 function could be disrupted in ft61 alleles; perhaps a more likely scenario is that Fat61 protein binds to Fbxl7 but cannot modulate its activity because it fails to recruit an accessory protein that either modulates Fbxl7 activity or that is regulated by Fbxl7*.

We have written a more conservative concluding sentence to this paragraph: “Thus, the apical localization of Fbxl7 requires the Ft D domain but neither the F domain nor the motif that is disrupted by the *ft*^*61*^ allele.”

*4) The text relating to the blots in*
Figure 6—figure supplement 1
*states that Dachs levels do not go up when Fbxl7 is increased but they actually do look higher. This also seems to hold true in*
Figure 7*. However, in*
Figure 7*, Fbxl7 RNAi does not seem to affect Dachs levels*.

We agree with the reviewers that Dachs levels go up slightly when Fbxl7 is overexpressed in S2 cells. We have added this observation to the text.

*5) The final model figure is not useful, and is at times misleading. Particularly, the labeling of the regions 1,2 and 3 is not helpful, as it adds yet another unmatching set of designations to the Ft cytoplasmic domain. It would be more useful if the authors adapted*
Figure 5
*to their final model. Also, the arrows to Dachs might be better if they were dotted, or via question marks, or possibly across cell membranes to Ds, as all the data in their paper indicates that the link to Dachs is indirect*.

Thank you for these suggestions. We have, as suggested by the reviewers, adapted Figure 5 to the final model and modified the arrows as requested.

*6) The manuscript gets a bit bogged down with the negative data about the possible type of vesicles, possible functions of Dachs and the inclusion of the Cindr data. The manuscript would be improved if this section of the paper was reduced, as it is largely negative data. Similarly, it would be more useful if the Discussion addressed what the authors have shown and their model, and reduce the discussion of vesicles and Cindr, as the link to the title and the Abstract is not carried through*.

While we agree with the reviewer that the data in this section do not provide a clear answer as to the direct target of Fbxl7, we also think that these experiments provide some of the best hints for where its direct targets may function (i.e. in vesicle trafficking). By showing these data, we hope that other scientists who work on vesicle trafficking will take notice of the localization of Fbxl7 and make connections with other observations that might point to candidate targets. We therefore think that it is important that these results are retained in the manuscript.